# Zero-Shot Off-Policy Learning

**Arip Asadulaev** [1*]   **Maksim Bobrin** [234*]   **Salem Lahlou** [1]   **Dmitry Dylov** [23]   **Fakhri Karray** [1]   **Martin Takac** [1]

## Abstract

Off-policy learning methods seek to derive an optimal policy directly from a fixed dataset of prior interactions. This objective presents significant challenges, primarily due to the inherent distributional shift and value function overestimation bias. These issues become even more noticeable in *zero-shot* reinforcement learning, where an agent trained on reward-free data must adapt to new tasks at test time without additional training. In this work, we address the off-policy problem in a zero-shot setting by discovering a theoretical connection of successor measures to stationary density ratios. Using this insight, our algorithm can infer optimal importance sampling ratios, effectively performing a stationary distribution correction with an optimal policy *for any task on the fly*. We benchmark our method in motion tracking tasks on `SMPL Humanoid`, continuous control on `ExoRL`, and for the long-horizon `OGBench` tasks. Our technique seamlessly integrates into forward-backward representation frameworks and enables *fast*-adaptation to new tasks in a *training-free* regime. More broadly, this work bridges off-policy learning and zero-shot adaptation, offering benefits to both research areas.

## 1. Introduction

Unsupervised (or self-supervised) pretraining has become a central ingredient behind recent progress in foundation models. In this approach, large, unlabeled datasets are leveraged to learn reusable structure, which is then specialized with lightweight supervision. Translating this recipe to Reinforcement Learning (RL) promises *generalist agents* that acquire behavioral priors from offline data and can be prompted for any task at deployment time. These agents are often referred to as *Behavioral Foundation Models (BFMs)*

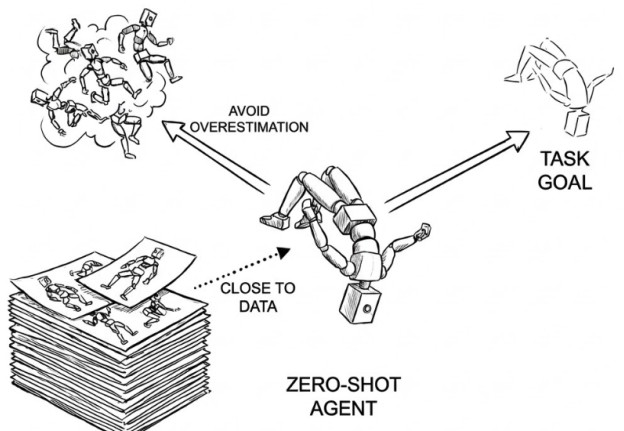

*Figure 1.* A sketch of the zero-shot off-policy adaptation. The agent completes the new task by combining actions from its experience.

([Park et al., 2024](); [Agarwal et al., 2025c]()).

In this *unsupervised zero-shot RL* paradigm, the model learns a representation of a large class of possible downstream tasks. Specifically, it learns a *family of policies* indexed or conditioned by a low-dimensional task descriptor ([Agarwal et al., 2025a]()). At test time, a task specification is used to *retrieve* a corresponding policy without additional learning or finetuning stage. Such specifications can be prompted via text ([Vainshtein et al., 2025]()), goals ([Tirinzoni et al.]()), motion capture ([Pirotta et al., 2023](); [Li et al., 2025]()), or videos ([Sikchi et al., 2024]()). However, this inference-based adaptation inherits a core difficulty of data-driven decision making. The policy induced by the inferred task can query parts of the state(-action) space that are poorly represented in the pretraining data. This leads to systematic errors in value estimates and, consequently, suboptimal behavior.

This perspective connects BFMs to the challenges of *off-policy* and *offline* RL ([Levine et al., 2020]()). Offline RL aims to synthesize new behavior by reusing previously collected experience. Yet a learned policy typically induces a visitation distribution that differs from the data-collecting behavior, so value estimation can become incorrect when the policy queries out-of-distribution (OOD) regions ([Kumar et al., 2020]()). Unlike offline RL where the objective is tied to a single task, BFMs must *generalize across tasks* that may only be revealed at test time, making robust off-policy reasoning even more critical (Fig. 1) ([Jeen et al., 2024]()).

---

[*]Equal contribution  [1]MBZUAI [2]Applied AI Institute, Computational Imaging Lab [3]AXXX [4]MBZUAI (Internship). Correspondence to: Arip Asadulaev <arip.asadulaev@mbzuai.ac.ae>.

*Proceedings of the $43^{rd}$ International Conference on Machine Learning*, Seoul, South Korea. PMLR 306, 2026. Copyright 2026 by the author(s).

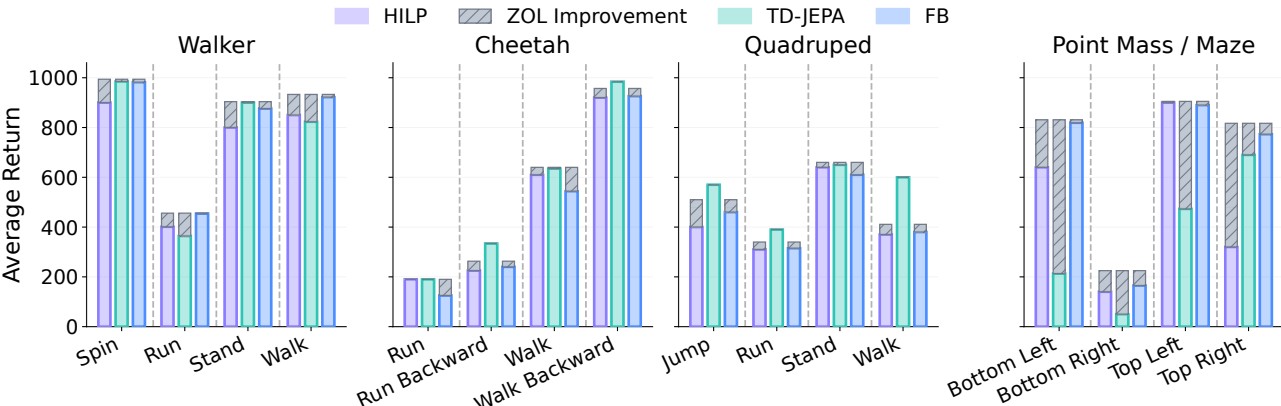

*Figure 2.* ZOL improvements over other BFMs on ExORL (DMC-state) benchmark. For full results, refer to Tbl. 2. Despite not having access to the environment, ZOL performs better, especially on long horizon tasks.

A principled way to reason about this mismatch is through *stationary occupancy correction*. Formally, let $d^\beta$ denote the discounted occupancy induced by the data-collecting behavior $\beta$, and $d^\pi$ the occupancy induced by a policy $\pi$. Their stationary density ratio $w_\pi = d^\pi/d^\beta$ quantifies how strongly $\pi$ shifts visitation relative to the dataset. The goal is to find an optimal $w^*$, that acts as the correction ratio with an *unknown, optimal* task-policy $d^{\pi^*}$. Off-policy evaluation and optimization via stationary occupancy correction have been studied by the DICE family of algorithms (Nachum et al., 2019a;b; Lee et al., 2021; Mao et al., 2024). However, these approaches require the solution of a task-specific optimization, which is not in accordance with the BFM-style zero-shot adaptation. Thus, we state the following question:

> *Motivation*: Can we estimate optimal stationary occupancy ratios for a whole family of BFM-induced policies without per-task online optimization, enabling fast test-time adaptation while avoiding the pathologies of off-policy learning?

We answer this question with a theoretical insight in Eq.(9): the Forward-Backward (FB) framework (Touati & Ollivier, 2021) for estimating Successor Measures (SMs) (Dayan, 1993) yields a bridge from behavior representations to *occupancy correction*. Concretely, the bilinear FB structure implicitly implies a low-rank parameterization of stationary density ratios *across the latent policy family* induced by the BFM, making it possible to compute $w_\pi$ in a zero-shot manner from pretrained components (see §3).

Building on this connection, we propose *Zero-Shot Off-policy Learning* (ZOL), a simple test-time procedure for *occupancy-aware fast adaptation*, without changing the training procedure of the BFMs. Given a small amount of task information (e.g., reward-labeled samples or sparse goals), ZOL uses the FB-induced ratio to (i) reweight the supervision toward dataset-supported regions and (ii) in-

fer a task latent that selects optimal behaviors consistent with the new objective while remaining grounded in the pretraining distribution. This targets the inference gap: ZOL biases adaptation away from latents that *look good under extrapolated values* but induce unsupported visitation, and toward latents that are *task-relevant* and *dataset-consistent*. As a result, ZOL mitigates overestimation and extrapolation errors during zero-shot adaptation, requires no additional training and fully offline, and comes with a theoretical justification.

Importantly, recent evidence suggests that even when a BFM contains strong behaviors in its latent policy space, the inference procedure itself can return policy representations that are far from optimal for the task (Sikchi et al., 2025; Bobrin et al., 2025). In other words, the performance gap may arise from *how* adaptation is performed, not only from *what* the pretrained model can express. Our method also contributes to addressing this problem by providing improved adaptation across diverse tasks while explicitly preferring policies whose visitation distributions remain supported by the pretraining data.

We evaluated ZOL on a diverse suite of zero-shot adaptation settings, including long horizon goal-conditioned control on OGBench, downstream tasks on ExORL, and MoCap imitation on AMASS, complemented by a simple 2D toy domain that isolates the core mechanism as a proof of concept (§5). Across all benchmarks, ZOL yields consistent gains in robustness and reliability over existing zero-shot adaptation methods and state-of-the-art BFMs (§6).

## 2. Background

### 2.1. Markov Decision Processes

Consider a discounted Markov decision process (MDP) $\mathcal{M} = \langle S, A, P, R, \gamma, \rho_0 \rangle$, where $S$ is the state space, $A$ is the action space, $P(\cdot \mid s, a) \in \Delta(S)$ is the transition

kernel, $R(s,a) \in \mathbb{R}$ is the reward function, $\gamma \in [0,1)$ is the discount factor and $\rho_0 \in \Delta(S)$ is the initial state distribution. If convenient, $\rho_0$ can be extended to an initial state-action distribution via $\rho_0(s,a) = \rho_0(s)\pi(a \mid s)$. A (stationary) policy $\pi(\cdot \mid s) \in \Delta(A)$ induces a trajectory distribution over $(s_t, a_t)_{t\geq0}$ through $a_t \sim \pi(\cdot \mid s_t)$ and $s_{t+1} \sim P(\cdot \mid s_t, a_t)$. A central object in our analysis is the *discounted stationary occupancy measure* of $\pi$:

$$d^\pi(s,a) := (1-\gamma)\sum_{t=0}^{\infty} \gamma^t \Pr(s_t = s,\ a_t = a \mid \pi) \quad (1)$$

Intuitively, $d^\pi(s,a)$ is the discounted frequency with which the pair $(s,a)$ is visited when starting from $\rho_0$ and following $\pi$. By construction, $d^\pi$ is a probability distribution and sums up to 1 over state-action pairs.

### 2.2. Offline RL and Stationary Density Ratios

In offline RL, we do not interact with $\mathcal{M}$ directly. Instead, we are given a static dataset $\mathcal{D} = \{(s_i, a_i, r_i, s_i')\}_{i=1}^N$ collected by an unknown behavior policy $\beta$ (Levine et al., 2020). We view $\mathcal{D}$ as being drawn from the behavior-induced occupancy $d^\beta$, i.e., $(s_i, a_i) \sim d^\beta$ and $s_i' \sim P(\cdot \mid s_i, a_i)$ (with rewards obtained from $R$ when available).[1] The fundamental difficulty in off-policy methods is in *distribution shift*: we seek to evaluate or optimize a target policy $\pi$ whose occupancy $d^\pi$ may differ substantially from $d^\beta$. A standard tool to relate these distributions is the *stationary density ratio* (a.k.a. stationary distribution correction):

$$w_{\pi/\beta}(s,a) := \frac{d^\pi(s,a)}{d^\beta(s,a)} \quad (2)$$

When $d^\beta(s,a) = 0$, the ratio is undefined; throughout, we adopt the convention $w_{\pi/\beta}(s,a) = 0$ in this case, which enforces a *support constraint* that forbids extrapolation beyond the dataset. While classical methods estimate $w_{\pi/\beta}$ for a single fixed $\pi$, more recent approaches aim to learn ratio representations that generalize across a family of target policies (Lee et al., 2021), a viewpoint that naturally aligns with BFMs, where adaptation amounts to selecting (or conditioning on) a policy from a large pretrained family.

### 2.3. Forward-Backward Successor Representations

The *successor measure* (SM) (Dayan, 1993; Blier et al., 2021), denoted $M^\pi$, characterizes the discounted future visitation induced by a policy $\pi$ starting from $(s_0, a_0)$:

$$M^\pi(s_0, a_0, s, a) := \sum_{t=0}^{\infty} \gamma^t \Pr(s_t = s,\ a_t = a \mid s_0, a_0, \pi) \quad (3)$$

---

[1] This i.i.d. assumption is standard for analysis; in practice, $\mathcal{D}$ may be temporally correlated, and $d^\beta$ should be interpreted as the dataset's empirical discounted visitation distribution.

Unlike the value function, which collapses future outcomes into a scalar, $M^\pi$ is reward-free and captures *where* the agent is expected to go in the future.

Let $d^\beta$ be a reference distribution (e.g., the dataset occupancy). We often express the SM as a density $m^\pi$ with respect to $d^\beta$:

$$M^\pi(s_0, a_0, s, a) = m^\pi(s_0, a_0, s, a)\, d^\beta(s,a) \quad (4)$$

Here, $m^\pi(s_0, a_0, s, a)$ can be interpreted as a discounted likelihood ratio of visiting $(s,a)$ in the future from $(s_0, a_0)$ under $\pi$, normalized by the prevalence of $(s,a)$ under the reference $d^\beta$. Because the SM separates dynamics from rewards, the $Q$-function admits a linear form:

$$Q_r^\pi(s_0, a_0) = \int M^\pi(s_0, a_0, s, a)\, r(s,a)\, \mathrm{d}s\, \mathrm{d}a \quad (5)$$

(with the integral replaced by a sum in discrete spaces).

**Forward–Backward factorization.** Successor measures are particularly useful for zero-shot transfer: once $M^\pi$ (or a representation of it) is learned, it can be reused across tasks by changing only the reward. The Forward–Backward (FB) framework (Touati & Ollivier, 2021) makes this tractable by approximating the successor density with a low-rank factorization conditioned on a latent variable $z \in \mathbb{R}^d$. Consider a family of policies $\{\pi_z\}$ indexed by $z$, and two learned embeddings: a forward embedding $F : S \times A \times \mathbb{R}^d \to \mathbb{R}^d$ and a backward embedding $B : S \times A \to \mathbb{R}^d$. FB approximates

$$m^{\pi_z}(s_0, a_0, s, a) \approx F(s_0, a_0, z)^\top B(s, a) \quad (6)$$

If the task reward can be represented linearly in the backward features, $r(s,a) \approx B(s,a)^\top z$, then combining Eqs.(5)-(6) yields the familiar FB score: $Q_r^{\pi_z}(s_0, a_0) \approx F(s_0, a_0, z)^\top z$ Consequently, a greedy policy for task latent $z$ is obtained by maximizing this score over actions:

$$\pi_z(s) \approx \arg\max_a\ F(s, a, z)^\top z \quad (7)$$

**Interpretation.** Intuitively, $F(\cdot, \cdot, z)$ predicts the expected discounted accumulation of backward features along trajectories induced by $\pi_z$ (successor features), while $B(\cdot, \cdot)$ provides a task-agnostic basis over state–action pairs (Barreto et al., 2017; Borsa et al., 2018; Agarwal et al., 2025b). The latent $z$ serves as a compact task embedding: it specifies both (approximately) the reward and the corresponding policy through Eq.(7). During training, sampling different $z$ values encourages diverse behaviors by exploring different directions in this feature space.

**Learning.** The FB embeddings are trained by enforcing a Bellman fixed-point for the successor density, $m^{\pi_z} = \mathbf{I} + \gamma P^{\pi_z} m^{\pi_z}$, together with the factorization in Eq. (6), via temporal-difference updates (Touati & Ollivier, 2021). Additional loss is also added to enforce $B^T B = I$ to enhance expressivity and avoid trivial solutions.

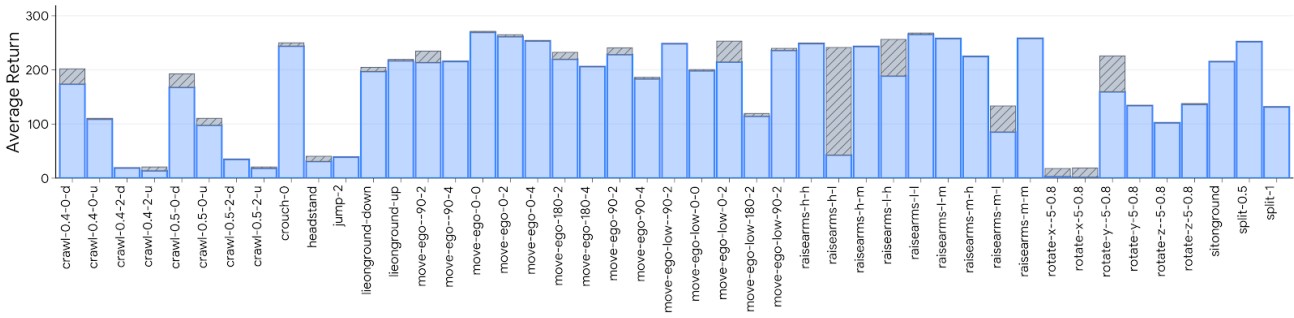

*Figure 3.* **HumEnv benchmark.** Performance improvement of ZOL over FB-CPR on SMPL Humanoid, regularized to diverse realistic motion capture trajectories from AMASS dataset.

**Test-time Inference & Policy Extraction.** At test time, given a new task specified by a reward $r_{\text{new}}(s, a)$, the goal is to infer a latent $z^\star$ that maximizes return. If $r_{\text{new}}$ can be approximated by the learned basis, e.g., $r_{\text{new}}(s, a) \approx B(s, a)^\top \mathbf{w}$, one can estimate a corresponding task embedding from the dataset:

$$z = \mathbb{E}_{(s,a)\sim\mathcal{D}}\big[B(s, a)\, r_{\text{new}}(s, a)\big]. \quad (8)$$

Recent studies ([Sikchi et al., 2025](); [Bobrin et al., 2025]()) report that the latent $z$ inferred by Eq. (8) can be systematically *suboptimal*. The issue is due to approximation errors during unsupervised training, which can steer $z$ toward a poor region of the latent space. This motivates augmenting zero-shot inference with an additional adaptation mechanism that can reliably *navigate* the latent policy family to uncover better task policies.

## 3. Method

In the current section, we show the exact connection between DICE correction and Forward-Backward successor representations. We show how occupancy ratios can be obtained directly from pretrained FB, yielding a simple low-rank estimator that can be evaluated *only on dataset samples*.

For any policy $\pi$, the discounted stationary occupancy $d^\pi$ can be written as an average of the successor measure over the initial distribution:

$$d^\pi(s, a) = (1 - \gamma)\, \mathbb{E}_{(s_0,a_0)\sim\rho_0}\left[M^\pi(s_0, a_0, s, a)\right]$$

Using the density decomposition in (4), i.e., $M^\pi(s_0, a_0, s, a) = m^\pi(s_0, a_0, s, a)\, d^\beta(s, a)$, we obtain

$$d^\pi(s, a) = (1 - \gamma)\, \mathbb{E}_{(s_0,a_0)\sim\rho_0}\left[m^\pi(s_0, a_0, s, a)\right] d^\beta(s, a).$$

Dividing by $d^\beta(s, a)$ yields the exact identity

$$w_{\pi/\beta}(s, a) = \frac{(1-\gamma)\, \mathbb{E}_{(s_0,a_0)\sim\rho_0}\left[m^\pi(s_0, a_0, s, a)\right] \cancel{d^\beta(s, a)}}{\cancel{d^\beta(s, a)}}$$

$$w_{\pi/\beta}(s, a) = (1 - \gamma)\, \mathbb{E}_{(s_0,a_0)\sim\rho_0}\left[m^\pi(s_0, a_0, s, a)\right] \quad (9)$$

Thus, stationary distribution correction is the $(1 - \gamma)$-scaled expected successor *density* under $\pi$, averaged over starting pairs $(s_0, a_0) \sim \rho_0$.

With FB, for a latent-indexed policy family $\{\pi_z\}$, the successor density is approximated by (6): $m^{\pi_z}(s_0, a_0, s, a) \approx F(s_0, a_0, z)^\top B(s, a)$. Taking the expectation over $(s_0, a_0) \sim \rho_0$ gives

$$\mathbb{E}_{\rho_0}\big[m^{\pi_z}(s_0, a_0, s, a)\big] \approx \mathbb{E}_{\rho_0}\big[F(s_0, a_0, z)^\top B(s, a)\big],$$

and since $B(s, a)$ depends only on $(s, a)$, it can be pulled outside the expectation. Define the *forward expectation vector*

$$W_{\pi_z} := \mathbb{E}_{(s_0,a_0)\sim\rho_0}\left[F(s_0, a_0, z)\right] \in \mathbb{R}^d$$

Substituting into (9) yields a compact low-rank estimator of the stationary density ratio:

$$\boxed{w_{\pi_z^*/\beta}(s, a) = \frac{d^{\pi_z^*}(s, a)}{d^\beta(s, a)} \;\approx\; (1 - \gamma)\, W_{\pi_z}^\top B(s, a)} \quad (10)$$

That is, for a given task/policy latent $z$, the ratio is linear in the backward embedding $B(s, a)$, with coefficients given by the expected forward embedding $W_{\pi_z}$. Intuitively, $W_{\pi_z}$ summarizes how the latent policy family shifts visitation from the dataset, while $B$ provides the state-action basis on which this shift is expressed. Moreover, Eq. (10) aims to find an optimal correction ratio between unknown, optimal policy for task $z$ and prior data $d^\beta$.

Crucially, Eq.(10) is only ever evaluated on in support $(s, a) \sim d^\beta$. This avoids extrapolating $w_{\pi/\beta}$ to unseen regions and does not require access to the behavior policy $\beta$ itself, only its induced occupancy through the dataset and the learned embedding $B(s, a)$.

> ***Takeaway:*** Forward–Backward representations implicitly contain the distribution-correction weights $w_{\pi/\beta}$, enabling a training-free approximation of occupancy-corrected (and, in our setting, near-optimal) adaptation for the task latents $z$ at test time.

### 3.1. Algorithm

Our method consists of three steps: (i) training the forward-backward representations, (ii) inferring a task-specific latent vector $z$, and (iii) optimizing this vector using our objective to find a better value for $z$. First, we train the forward ($F_\theta$) and backward ($B_\psi$) embedding networks by minimizing the FB temporal-difference objective. Given a policy $\pi(\cdot \mid s, z)$ and a latent distribution $\nu(z)$, we minimize the loss:

$$\delta \triangleq F_\theta(s, a, z)^\top B_\psi(s^+) - \gamma \, \bar{F}(s', a', z)^\top \bar{B}(s^+),$$

$$\mathcal{L}_{\text{FB}} = \mathbb{E}\big[\delta^2 - 2 F_\theta(s, a, z)^\top B_\psi(s')\big]$$

Second, for a new task $T$ with reward $r_T$ and dataset $\mathcal{D}$, we infer a task-specific embedding $c_T$. This embedding aggregates reward information via the backward features and serves as the initialization for $z$:

$$c_T := \mathbb{E}_{(s,a)\sim\mathcal{D}}\big[B_\psi(s, a)\, r_T(s, a)\big]. \tag{11}$$

Finally, we optimize the latent variable $z$ to maximize the task return. Using the FB approximation, our optimization objective function is given by (10). We perform a gradient ascent (using Adam optimizer (Kingma & Ba, 2014) in our experiments) in the space of $z$. We estimate the successor feature expectation $W_{\pi_z} = \mathbb{E}_{s_0 \sim \rho, \pi_z}[F(s, a, z)]$ via Monte Carlo sampling. In each optimization step, we sample a batch of start states $s_0$ from the environment, execute the policy $\pi(\cdot|s_0, z)$, and average the resulting embeddings $F$. In practice, to ensure stable optimization, we implement:

**Positive Density Ratios.** To guarantee valid density ratios, we use the Softplus activation $\sigma(x) = \log(1+e^x)$ in Eq.(10). The weights are also normalized over the batch $\mathbb{E}[w] \approx 1$.

**Regularized Objective.** We maximize a penalized objective $\mathcal{J}_{\text{total}} = J_{\text{ret}} - \lambda_1\chi^2 - \lambda_2\mathcal{L}_{\text{trust}}$. Here, $J_{\text{ret}}$ is the weighted return using centered rewards ($r \leftarrow r - \bar{r}$) to prevent trivial solutions. The $\chi^2$ term, defined as $\mathbb{E}[(w - 1)^2]$, penalizes high variance in the importance weights. The trust region term $\mathcal{L}_{\text{trust}} = \|\text{proj}(z) - \text{proj}(z_{\text{init}})\|^2$ ensures that the latent search stays within the region of initial task. We also apply correction ration clipping to prevent numerical instability. We provide ablation study over the all of these parameters in (§7). For the theoretical considerations which support above choices, we refer to Appendix C.

In total, we use this ratio to estimate the return of $\pi_z$ without online rollouts via regularized weighted-return objective:

$$J_{\text{total}}(z) = J(\pi_z) - \lambda_1\chi^2(z) - \lambda_2\mathcal{L}_{\text{trust}}(z) \tag{12}$$

where $J_{\text{ret}}(\pi_z) = \mathbb{E}_{(s,a,r)\sim d^\beta}\big[w_{\pi_z/\beta}(s, a)\,(r(s, a) - \bar{r})\big]$.

Thus, we do not maximize the density ratio itself. We maximize an offline estimate of return with the ratio correcting the mismatch between $d^{\pi_z}$ and $d^\beta$. The method is conservative, evaluating only on $d^\beta$ support, with $w_{\pi_z/\beta}(s, a) = 0$ when $d^\beta(s, a) = 0$. For the exact ratio, $\mathbb{E}_{d^\beta}\big[w_{\pi_z/\beta}\big] = 1$, so the goal is not to make the ratio uniformly small, but to use it as a valid occupancy correction. Stability of search over $z$ is ensured via positivity, normalization, clipping, and trust-regions. We will revise Sec. 3.1 to make this clear.

## 4. Related Work

Stationary occupancy density ratios are at the core of many offline RL algorithms. For example, in DualDICE (Nachum et al., 2019a) and OptiDICE (Lee et al., 2021), $w_{\pi/\beta}$ is estimated by solving a saddle-point optimization problem. Diffusion-DICE (Mao et al., 2024) views distribution correction in the context of iterative policy improvement via diffusion guidance. All of those methods involve solving high-dimensional optimization problems over function classes that directly map $(s, a)$ to a scalar density ratio. However, these methods are not applicable for zero-shot settings.

**ReLA** (Sikchi et al., 2025) performs adaptation online by *reward projection error* $r - \phi^\top z_r$ via the decomposition

$$Q_r^{\pi_z}(s, a) = F(s, a, z)^\top z_r + Q_{r-\phi^\top z_r}^{\pi_z}(s, a)$$

learning a residual critic $Q_{\text{res}}(s, a; \theta) \approx Q_{r-\phi^\top z_r}^{\pi_z}(s, a)$ while keeping the base term $\psi(\cdot)^\top z_r$ frozen. Latent adaptation starts from $z = z_r$ and updates $z$ using

$$\nabla_z \, \mathbb{E}_{s\sim D_{\text{online}}}\Big[F(s, \pi_z(s), z_r)^\top z_r + Q_{\text{res}}(s, \pi_z(s); \theta)\Big]$$

ReLA requires environment interaction to populate $D_{\text{online}}$ and depends on *rollout-collection* to approximate the on-policy improvement direction.

**LoLA** (Sikchi et al., 2025) optimizes a latent distribution $z \sim \mathcal{N}(\mu, \sigma)$ (initialized at $z_r$) using fixed-horizon on-policy rollouts and a frozen terminal bootstrap:

$$R_n(s_0, z) = \sum_{t=0}^{n-1} \gamma^t r(s_{t+1}) + \gamma^n F(s_{n+1}, \pi_z(s_{n+1}), z)^\top z_r$$

A REINFORCE-style (Williams, 1992) estimator (often with a baseline $b$) gives

$$\nabla_{\mu,\sigma} \, \mathbb{E}[R_n(s_0, z)] \approx \mathbb{E}\big[(R_n(s_0, z) - b)\, \nabla_{\mu,\sigma} \log p_{\mu,\sigma}(z)\big]$$

Approximating the *true on-policy* gradient direction fundamentally depends on Monte-Carlo rollouts and needs to manage a tradeoff: larger $n$ is closer to on-policy, but increases interaction cost and variance. LoLA also assumes the ability to reset the environment to replay-buffer.

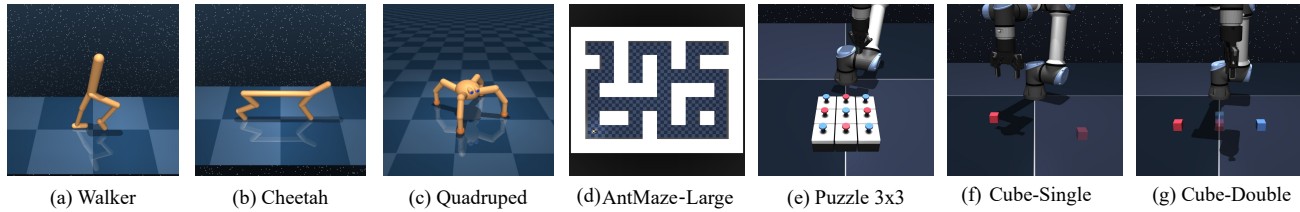

(a) Walker     (b) Cheetah     (c) Quadruped     (d)AntMaze-Large     (e) Puzzle 3x3     (f) Cube-Single     (g) Cube-Double

*Figure 4.* **Evaluation Benchmarks.** We evaluate ZOL on seven robotic locomotion and manipulation environments across 5 different tasks in unsupervised regime, which vary in complexity, data coverage, and reward types.

> *Takeaway:* Classic DICE methods are not applicable for zero-shot settings. ReLA/LoLA require test-time environment access and are based on *online*, on-policy rollouts, possessing a trade-off.

In contrast, ZOL performs training-free, interaction-free test-time adaptation by applying distribution/occupancy correction. Our correction recovers the on-policy objective/gradient *without* requiring MC rollouts, yielding a behavior-supported update that mitigates OOD *greedification*.

## 5. Sub-optimality of Zero-Shot Adaptation

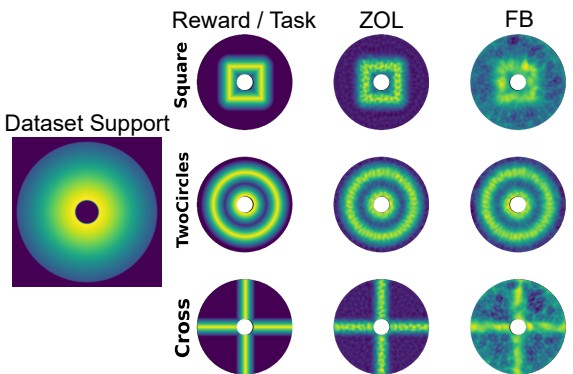

*Figure 5.* The offline dataset support (left), and the corresponding heatmaps of the reconstructed rewards from latents obtained by ZOL (third column) or FB (fourth column). Colors represent rewards (higher → brighter).

We now construct a minimal 2D example with the goal to isolate and visualize a failure mode of *inference-time* latent selection in FB( Eq. (8)). States are points $s = (x, y) \in \mathbb{R}^2$ constrained to an *donut* with inner radius $rad_{\min} = 0.25$ and outer radius $rad_{\max} = 1.5$. We pretrain FB purely from an offline transitions dataset on this donut, collected by a random-walk behavior policy (uniform actions clipped to $[-0.1, 0.1]^2$) and *without* any task reward labels. The learned backward map $B$ and forward map $F(s, a, z)$ therefore summarize *unsupervised* structure of the dynamics under the dataset across various tasks encodings $z$.

**Offline support distribution.** A key ingredient in this toy is that the offline data provides a *biased support* over states.

Moreover, we enforce the dataset support to have a radially-decaying density restricted to the donut:

$$d^\beta(s) \; \propto \; \exp\Big( - \tfrac{\|s\|_2^2}{2\sigma^2}\Big) \cdot \mathbb{1}\{rad_{\min} \leq \|s\|_2 \leq rad_{\max}\}$$

so that most mass lies closer to the inner region of the annulus (leftmost panel in Fig. 5). Intuitively, this mimics the common offline-RL regime where the dataset covers some region densely while only sparsely covering other regions that may be crucial for downstream tasks.

**Downstream Tasks.** At test time, we consider three reward functions, acting as different tasks, defined over the donut: SQUARE, TWOCIRCLES, and CROSS (second column in Fig. 5). For each task, we visualize the reconstructed rewards for each state as $r_z(s) = B(s)z$, where $z$ is obtained either from ZOL latent optimization procedure or through FB.

Given a batch of offline states $\{s_i\}_{i=1}^N \sim d^\beta$ and corresponding reward labels $\{r_i = r(s_i)\}$ for certain task, FB chooses $z$ which is dominated by explaining $r(s)$ *where data density is high*, even if the task-critical structure lies in low-support regions. Importantly, this is not a representational limitation of FB and failure stems from the *suboptimality of the inference step under support bias*, i.e., selecting $z$ using an objective that is dominated by $d^\beta$.

Our approach (ZOL) replaces the purely support-conditioned latent selection with an inference objective that explicitly reasons about the *state visitation induced by the candidate latent*. Concretely, for a candidate $z$ we estimate a (successor) occupancy score over states using the FB model, and then form normalized weights (stationary correction) that upweight states that are more likely under the policy induced by $z$.

**Takeaway.** Fig. 5 shows that ZOL reliably reconstructs the intended task structure (third column, ZOL) across all three rewards, recovering sharp reward structure. In contrast, vanilla FB inference (fourth column) systematically collapses toward smoother, support-biased explanations, failing to reconstruct task representation properly. Overall, this toy highlights the core insight: even when the unsupervised representation is sufficiently expressive, *zero-shot adaptation can be suboptimal if latent inference is performed solely*

*Table 1.* Average results across 5 tasks on the OGBench comparing ZOL against other approaches FB and on-policy (LoLA/ReLA).

| TASK | ReLA | LoLA | FB | **ZOL** |
|---|---|---|---|---|
| ANTMAZE-L-NAVIGATE | **0.77** ±0.1 | 0.6 ±0.2 | 0.68 ±0.2 | 0.63 ±0.1 |
| ANTMAZE-L-STITCH | 0.2 ±0.2 | 0.3 ±0.2 | 0.36 ±0.2 | **0.3** ±0.1 |
| ANTMAZE-M-STITCH | 0.6 ±0.3 | 0.7 ±0.3 | 0.69 ±0.3 | **0.73** ±0.2 |
| ANTMAZE-M-EXPLORE | 0.4 ±0.3 | 0.6 ±0.3 | 0.62 ±0.2 | **0.67** ±0.3 |
| ANTMAZE-M-NAVIGATE | 0.8 ±0.2 | 0.9 ±0.1 | **0.95** ±0.0 | 0.91 ±0.1 |
| CUBE-DOUBLE-PLAY | **0.03** ±0.0 | 0.0 ±0.0 | 0.0 ±0.0 | 0.01 ±0.0 |
| CUBE-SINGLE-PLAY | 0.26 ±0.1 | 0.3 ±0.1 | 0.31 ±0.1 | **0.34** ±0.0 |
| PUZZLE-3X3-PLAY | 0.07 ±0.1 | 0.0 ±0.0 | 0.0 ±0.0 | **0.14** ±0.2 |
| AVG. | 0.39 | 0.42 | 0.45 | **0.47** |

*through empirical correlations on a biased offline support.* By making inference self-consistent with the candidate policy's induced occupancy, ZOL mitigates this failure mode and yields substantially better zero-shot adaptation.

# 6. Experiments

We evaluate our approach on three complementary benchmarks designed to assess test-time adaptation under varying degrees of reward sparsity, task horizons, and state-space dimensionality. These environments probe an agent's capability to leverage task-agnostic pretraining data to solve specific downstream tasks.

For the baselines, we take most recent zero-shot RL approaches: FB (Touati & Ollivier, 2021), HILP (Park et al., 2024), TD-JEPA (Bagatella et al., 2025). For other baselines, which focus on the same setting of fast adaptation over pretrained BFM at test time, we take LoLA and ReLA (Sikchi et al., 2025), which described in §4. For the Humanoid, we compare to pretrained FB-CPR (Tirinzoni et al.). Detailed descriptions of baselines and hyperparameter configurations are provided in Appendices A and D.

## 6.1. ExORL DeepMind Control

We evaluate the zero-shot capabilities of ZOL on the ExORL benchmark (Yarats et al., 2022), a rigorous testbed for evaluating Behavioral Foundation Models (BFMs) under fixed exploratory data. Following the protocol in Laskin et al. (2021), we pretrain all baselines on the RND (Burda et al., 2019) exploratory dataset across four diverse domains: Walker, Cheetah, Quadruped, and Pointmass. This dataset is characterized by a lack of task-specific rewards, while being high coverage across state-action pairs, requiring the model to capture the global transition dynamics and state-occupancy measures.

**Evaluation Protocol.** We consider 16 downstream continuous tasks in total. For each task, we perform ZOL latent policy search for specific number $N$ of iterations (see Appendix D for the details) to identify the directional movement in the latent space that maximizes the inner product with the occupancy ratio correction. To ensure statistical significance, we report the mean and standard deviation

of unnormalized returns across 100 independent test-time rollouts per task. As shown in Table 2, ZOL consistently outperforms recent promptable baselines, particularly in high-dimensional domains like Quadruped.

## 6.2. OGBench

To test the limits of our approach on long-horizon reasoning and high-dimensional manipulation, we utilize OGBench (Park et al., 2025). This benchmark is significantly more challenging than DMC due to its sparse reward structure and the requirement for temporal stitching. We evaluate on five domains: antmaze-medium, antmaze-large, cube-double, cube-single, and puzzle-3x3. For the AntMaze environments, we specifically focus on the stitch and explore splits, which contain disjoint trajectories that must be connected via the learned Hilbert representation to reach the goal.

**Inference vs. Finetuning.** Our method achieves superior success rates (Table 1) without a single step of additional environment sampling, unlike LoLA and ReLA (Sikchi et al., 2025). This result empirically validates our hypothesis that the distribution correction ratio is sufficient to recover near-optimal policies, effectively bypassing the need for expensive test-time fine-tuning.

## 6.3. SMPL Humanoid

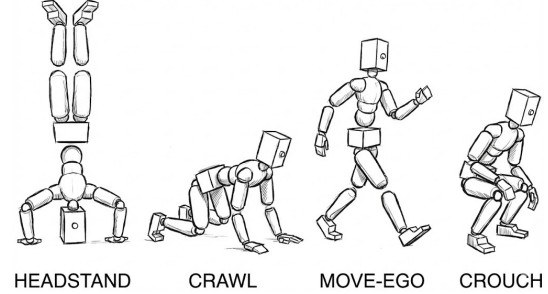

*Figure 6.* Illustration of tasks on SMLP Humanoid.

In this benchmark, we use the pretrained FB-CPR model of (Tirinzoni et al.) for HumEnv. The state is 358-dimensional (joint positions) and the action is 69-dimensional. We follow the model's inference recipe to infer $z_r$ via weighted regression on samples from an online buffer (using large replay batches from training stage), with buffer size $\sim$ 500k. For motion tracking, the latent is *time-varying*: we infer a separate $z_t$ at each step by averaging backward embeddings over a short future window, $z_t \propto \sum_{j=t+1}^{t+L+1} B_\theta(s_j)$, where $L$ matches the pretraining encoding length (Tirinzoni et al.). We evaluate with 50 rollouts and report mean return. This benchmark directly supports our DICE-style view of ZOL. FB-CPR pretraining regularizes the *joint* $(s, z)$ distribution induced by $\pi_z$ toward the AMASS-induced joint distribution, shaping a latent space of human-like skills (Fig. 6). At test time, prompting selects a region of this space, and

*Table 2.* **Average results on the state-based ExORL benchmark comparing ZOL against other approaches (HILP, FB, LoLA, ReLA, TD-JEPA).** The table reports the unnormalized return averaged over 100 rollouts on RND exploration dataset.

| ENVIRONMENT | TASK | HILP | FB | LoLA | ReLA | TD-JEPA | ZOL (OURS) |
|---|---|---|---|---|---|---|---|
| WALKER | SPIN | $900 \pm 5$ | $982 \pm 4$ | $980 \pm 2$ | $942 \pm 37$ | $985 \pm 6$ | $\mathbf{994} \pm 2$ |
| | RUN | $401 \pm 4$ | $454 \pm 3$ | $481 \pm 2$ | $382 \pm 3$ | $364 \pm 20$ | $\mathbf{456} \pm 2$ |
| | STAND | $800 \pm 6$ | $876 \pm 16$ | $828 \pm 2$ | $769 \pm 34$ | $900 \pm 18$ | $\mathbf{904} \pm 5$ |
| | WALK | $850 \pm 2$ | $922 \pm 3$ | $884 \pm 4$ | $590 \pm 8$ | $823 \pm 30$ | $\mathbf{933} \pm 2$ |
| CHEETAH | RUN | $190 \pm 5$ | $125 \pm 4$ | $142 \pm 6$ | $116 \pm 5$ | $190 \pm 50$ | $\mathbf{190} \pm 3$ |
| | RUN BACKWARD | $225 \pm 5$ | $240 \pm 5$ | $190 \pm 12$ | $245 \pm 6$ | $\mathbf{334} \pm 6$ | $263 \pm 4$ |
| | WALK | $610 \pm 6$ | $544 \pm 14$ | $801 \pm 29$ | $469 \pm 16$ | $635 \pm 50$ | $\mathbf{640} \pm 5$ |
| | WALK BACKWARD | $920 \pm 8$ | $926 \pm 14$ | $918 \pm 22$ | $961 \pm 8$ | $\mathbf{984} \pm 1$ | $957 \pm 5$ |
| QUADRUPED | JUMP | $400 \pm 10$ | $460 \pm 14$ | $558 \pm 8$ | $566 \pm 40$ | $\mathbf{570} \pm 30$ | $510 \pm 5$ |
| | RUN | $310 \pm 6$ | $315 \pm 7$ | $310 \pm 7$ | $437 \pm 12$ | $\mathbf{390} \pm 20$ | $340 \pm 3$ |
| | STAND | $640 \pm 5$ | $610 \pm 12$ | $640 \pm 7$ | $846 \pm 37$ | $650 \pm 45$ | $\mathbf{660} \pm 4$ |
| | WALK | $370 \pm 10$ | $380 \pm 9$ | $414 \pm 10$ | $417 \pm 10$ | $\mathbf{600} \pm 50$ | $411 \pm 3$ |
| POINT MASS / MAZE | BOTTOM LEFT | $640 \pm 5$ | $819 \pm 4$ | $804 \pm 4$ | $819 \pm 4$ | $213 \pm 10$ | $\mathbf{831} \pm 2$ |
| | BOTTOM RIGHT | $140 \pm 5$ | $165 \pm 2$ | $182 \pm 2$ | $132 \pm 2$ | $50 \pm 6$ | $\mathbf{225} \pm 1$ |
| | TOP LEFT | $900 \pm 4$ | $890 \pm 3$ | $931 \pm 4$ | $860 \pm 4$ | $473 \pm 2$ | $\mathbf{905} \pm 2$ |
| | TOP RIGHT | $320 \pm 15$ | $773 \pm 4$ | $807 \pm 4$ | $821 \pm 4$ | $690 \pm 60$ | $\mathbf{817} \pm 3$ |
| AVERAGE | | **538.5** | **592.5** | **616.9** | **585.8** | **553.2** | **627.2** |

distribution/occupancy correction addresses the mismatch between where the prompt is evaluated and where the rollout places mass. Figure 3 shows gains over baselines: ZOL finds substantially better policies. Full results are in Table 7.

## 7. Ablation Study

We systematically vary key ZOL hyperparameters to evaluate robustness and sensitivity, and find that performance is largely agnostic to these choices. Across the tested ranges,

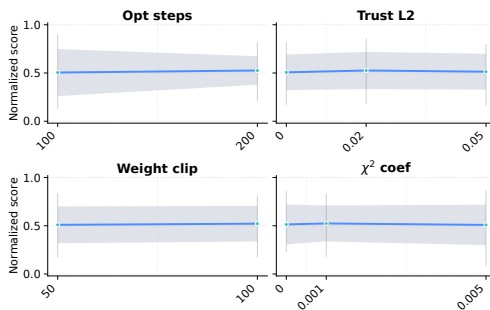

*Figure 7.* Hyperparameters ablation on the normalized return on the DMC benchmark.

ZOL is broadly robust: varying any single hyperparameter yields only modest changes in mean performance, indicating limited sensitivity and little need for extensive tuning. Performance consistently favors *fewer* inner optimization steps, suggesting diminishing returns and slight degradation consistent with over-optimization when the adaptation objective is optimized too aggressively. Regularization shows the expected pattern: a *small* trust penalty improves stability and typically increases return, whereas a large $\chi^2$ penalty becomes overly conservative and can reduce performance. Weight clipping is largely insensitive within this range and primarily functions as a safety constraint. The same results

were obtained on the SMPL Humanoid. Please see Fig.7 and 8, gray area depicts standard deviation across 100 rollouts. See Appendix F for details and SMPL ablation.

## 8. Conclusion & Limitations

We revisit fast adaptation for BFMs via stationary occupancy correction ratios. Although zero-shot task inference should, in principle, yield optimal policies, in practice it can produce suboptimal behaviors by choosing unrelated task encodings and inducing shift in the recovered policy. We link stationary density ratios to successor measures and show FB representations give a low-rank, data-only occupancy-correction estimator. Building on this, we propose *Zero-Shot Off-policy Learning (ZOL)*, a training-free test-time method that uses FB-induced ratios to reweight task supervision and steer latent policy selection toward task-aligned, dataset-consistent behavior. ZOL improves robustness over prior zero-shot baselines across benchmarks without changing pretraining or requiring online interaction.

**Limitations and future work.** ZOL inherits the representational bias of the learned FB factorization: when the ratio estimate is inaccurate or the downstream task requires sustained out-of-support behavior, correction can only trade off conservatism and performance. Promising directions include uncertainty-aware occupancy correction, stronger representations and richer ratio parameterizations, and extensions to stochastic, partially observable domains.

## Impact Statement

This paper presents work whose goal is to advance the field of Machine Learning. There are many potential societal consequences of our work, none which we feel must be specifically highlighted here.

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

*Table 3.* **FB vs. ZOL comparison on ExORL (state-based) with** 100 **evaluation rollouts.** We report mean $\pm$ std over 100 rollouts for FB and ZOL, and the absolute improvement $\Delta$ with relative gain.

| ENVIRONMENT (DMC-STATE) | TASK | FB | ZOL (OURS) | $\Delta$(IMPROVEMENT) |
|---|---|---|---|---|
| WALKER | SPIN | 982 ±2 | **994** ±2 | **+2.07** (0.21%) |
| | RUN | 454 ±3 | **456** ±2 | **+2.34** (0.51%) |
| | STAND | 876 ±16 | **904** ±5 | **+27.88** (3.18%) |
| | WALK | 922 ±3 | **933** ±2 | **+10.97** (1.19%) |
| CHEETAH | RUN | 125 ±4 | **190** ±3 | **+64.90** (51.79%) |
| | RUN BACKWARD | 240 ±5 | **263** ±4 | **+18.06** (7.35%) |
| | WALK | 544 ±14 | **640** ±5 | **+93.16** (17.12%) |
| | WALK BACKWARD | 926 ±14 | **957** ±6 | **+31.23** (3.37%) |
| QUADRUPED | JUMP | 460 ±14 | **510** ±5 | **+40.91** (8.71%) |
| | RUN | 315 ±7 | **345** ±3 | **+23.97** (7.61%) |
| | STAND | 610 ±12 | **660** ±4 | **+47.74** (7.79%) |
| | WALK | 380 ±10 | **421** ±9 | **+27.63** (7.19%) |
| POINT MASS / MAZE | BOTTOM LEFT | 819 ±4 | **831** ±2 | **+11.82** (1.44%) |
| | BOTTOM RIGHT | 165 ±2 | **225** ±1 | **+59.56** (35.98%) |
| | TOP LEFT | 890 ±3 | **905** ±2 | **+12.45** (1.40%) |
| | TOP RIGHT | 773 ±4 | **817** ±3 | **+43.36** (5.60%) |
| AVERAGE (ALL TASKS) | | **592.5** | **627.2** | **+32.38** (10.03%) |

## A. Related Works

**Behavioral foundation models.** *A behavioral foundation model* is an agent that, for a fixed MDP, can be trained unsupervised on reward-free transition data and then, at test time, immediately output approximately optimal policies for a broad family of reward functions $r$ without any extra learning or planning. In this work, we study zero-shot RL agents built on successor features and forward–backward representations. Below we outline most successful examples of BFMs.

**Universal Successor Features (USF)** (Borsa et al., 2018) methods learn *successor features* (SFs) that decouple dynamics from rewards by representing a policy through the expected discounted visitation of a feature map $\phi(s, a)$. Reward functions are assumed to be linear in features, $r(s, a) = w^\top \phi(s, a)$, so that values transfer by a simple inner product $Q^w(s, a) = w^\top \psi(s, a)$, where $\psi$ denotes the learned SF. After reward-free pretraining, a new task is specified by $w$, and the agent can act zero-shot by evaluating $w^\top \psi(s, a)$ and selecting actions accordingly; policy improvement can be obtained via generalized policy improvement (GPI) over a set of learned policies.

**HILP** (Park et al., 2024) HILP (Hilbert foundation policies) pre-trains a general latent-conditioned policy from unlabeled offline trajectories in two stages. First, it learns a Hilbert representation $\phi : S \to Z$ such that Euclidean distances in latent space approximate optimal temporal distances in the MDP, i.e., $d^*(s, g) \approx \|\phi(s) - \phi(g)\|$, by training a goal-conditioned value function parameterized as $V(s, g) = -\|\phi(s) - \phi(g)\|$ with an offline goal-conditioned value-learning objective. Second, it trains a latent-conditioned policy $\pi(a \mid s, z)$ (with $z$ sampled uniformly from the unit sphere) using offline RL on an intrinsic directional reward $r(s, z, s') = \langle \phi(s') - \phi(s), z \rangle$, so that maximizing reward for all directions $z$ yields a diverse set of long-horizon behaviors that span the latent space. At test time, the same policy can be "prompted" by choosing $z$ from a goal direction (e.g., $z \propto \phi(g) - \phi(s)$ for goal reaching) or fit from an arbitrary reward via linear regression against $\phi(s') - \phi(s)$ for zero-shot RL.

**TD-JEPA** (Bagatella et al., 2025) extends joint-embedding predictive objectives to RL by learning latent states that are *predictive under temporal-difference learning*. From reward-free transitions, it trains an encoder and a predictive model so that the embedding of a future state can be predicted from the current embedding and action, with TD-style bootstrapping providing a stable long-horizon learning signal. The resulting latent space is shaped to preserve controllable, temporally-extended structure, allowing downstream rewards to be handled by simple heads in the learned space. Consequently, once a reward function is specified, a policy can be produced zero-shot by evaluating the reward in the learned representation and acting greedily (or with a lightweight improvement step) without further environment interaction.

**FB-CPR.** (Tirinzoni et al.) learns a latent-conditioned policy family $\pi(\cdot \mid s, z)$ using Forward–Backward (FB) representa-

tions, and grounds this unsupervised policy space with an unlabeled behavior dataset of state-only trajectories. It embeds each demonstration trajectory $\tau$ into the same latent space as the policies using an FB-based encoder $z = \mathrm{ERFB}(\tau)$, which induces a joint distribution $p_{\mathcal{M}}(s, z)$ over states and inferred latents. Training regularizes FB policy improvement by matching the policy-induced joint distribution $p_\pi(s, z) = \nu(z)\rho^{\pi_z}(s)$ to $p_{\mathcal{M}}(s, z)$ via a KL penalty. To make this matching tractable, FB-CPR trains a latent-conditional discriminator $D(s, z)$ to estimate the density ratio between dataset samples $(s, \mathrm{ERFB}(\tau))$ and policy rollouts $(s, z)$, yielding a reward surrogate $\log \frac{p_{\mathcal{M}}(s,z)}{p_\pi(s,z)} \approx \log \frac{D(s,z)}{1-D(s,z)}$. A critic $Q(s, a, z)$ is learned off-policy to predict discounted return under this reward, and the actor is updated to maximize both the FB value term $F(s, a, z)^\top z$ and the regularization term $\alpha Q(s, a, z)$, interleaving environment rollouts with off-policy updates of $F, B, D, Q,$ and $\pi$.

For the hyperparameters of each of the methods we used in our experiments, see Appendix D.

## B. Analysis

## C. ZOL Algorithm Details & Proofs

### C.1. Setting and Notation

We assume an offline dataset (or replay batch) of $B$ state samples (or state–action samples)

$$\mathcal{D} = \{(s_i, r_i)\}_{i=1}^B,$$

collected by a behavior policy $\beta$, inducing a discounted occupancy distribution $d^\beta$. We also assume a pretrained Forward–Backward (FB) representation model with:

- a *backward embedding* $B_\theta(s) \in \mathbb{R}^d$ (often written $B(s, a)$; in many implementations it is state-only),
- a *forward embedding* $F_\theta(s, a, z) \in \mathbb{R}^d$ for latent $z \in \mathbb{R}^{d_z}$,
- a latent-indexed policy $\pi_z(a \mid s)$ from which we can compute an action $a = \pi_z(s)$ (in code: `model.act`).

The algorithm seeks an improved latent $z$ starting from a zero-shot (or inferred) latent $z_0$.

### C.2. FB-Induced Stationary Density Ratio Surrogate

A key theoretical identity in FB-based off-policy adaptation is that the stationary density ratio

$$w_z(s) \approx (1 - \gamma)\,\mu(z)^\top B_\theta(s),$$

where $\gamma \in (0, 1)$ is the discount and

$$\mu(z) := \mathbb{E}_{(s_0, a_0) \sim \rho_0}\big[F_\theta(s_0, a_0, z)\big] \in \mathbb{R}^d$$

is the expected forward embedding under the initial distribution $\rho_0$ and policy induced by $z$.

**Practical approximation of $\mu(z)$.** We approximate $\mu(z)$ by caching $N$ reset states $\{s_0^{(j)}\}_{j=1}^N$ and computing

$$\widehat{\mu}(z) = \frac{1}{N} \sum_{j=1}^N F_\theta\big(s_0^{(j)}, a_0^{(j)}, z\big), \quad a_0^{(j)} = \pi_z(s_0^{(j)}).$$

This estimate is differentiable with respect to $z$ because $z$ influences both $\pi_z$ and $F_\theta$.

### C.3. Weight Shaping: Positivity and Self-Normalization

Because function approximation can yield negative or arbitrarily scaled ratios, we shape the raw logits

$$\ell_i(z) := (1 - \gamma)\,B_\theta(s_i)^\top \widehat{\mu}(z)$$

into a stable positive weight. The algorithm uses:

1. **Positivity** via softplus:
$$\widetilde{w}_i(z) \;=\; \mathrm{softplus}\big(\ell_i(z)\big)$$

   ensuring $\widetilde{w}_i(z) > 0$.

2. **Self-normalization** to enforce $\mathbb{E}_{d^\beta}[w] \approx 1$:

$$w_i(z) \;=\; \frac{\widetilde{w}_i(z)}{\frac{1}{B}\sum_{j=1}^{B}\widetilde{w}_j(z)+\varepsilon},$$

   where $\varepsilon > 0$ prevents division by zero.

3. **Clipping** (optional) to prevent extreme weights:

$$w_i(z) \leftarrow \min\{w_i(z),\, w_{\max}\}.$$

### C.4. Reward Centering (Advantage-Style Baseline)

A frequent failure mode in offline optimization is collapsing to *do nothing* behaviors when rewards contain large constant offsets (e.g., an *alive bonus*). To mitigate this, the algorithm centers rewards:

$$r'_i \;=\; r_i - \frac{1}{B}\sum_{j=1}^{B} r_j.$$

This converts the objective into an *advantage-weighted* form that discourages trivial stationary solutions.

**Theorem C.1** (Centered reweighting equals the target policy return up to a constant)**.** *We write $w_\pi$ as an shorthand of $w_{\pi/\beta}$. Assume $d_\beta(s) > 0$ whenever $d_\pi(s) > 0$ and define the density ratio*

$$w_\pi(s) \;:=\; \frac{d_\pi(s)}{d_\beta(s)}.$$

*Then the centered reweighted objective*

$$J_c(\pi) \;:=\; \mathbb{E}_{s\sim d_\beta}\Big[w_\pi(s)\big(r(s)-\bar{r}_\beta\big)\Big] \quad \text{with} \quad \bar{r}_\beta := \mathbb{E}_{s\sim d_\beta}[r(s)]$$

*satisfies*

$$J_c(\pi) \;=\; \mathbb{E}_{s\sim d_\pi}[r(s)] \;-\; \bar{r}_\beta.$$

*Hence $\arg\max_\pi J_c(\pi) = \arg\max_\pi \mathbb{E}_{d_\pi}[r(s)]$.*

*Proof.* We have

$$\mathbb{E}_{d_\beta}[w_\pi(s)\,r(s)] = \sum_s d_\beta(s)\frac{d_\pi(s)}{d_\beta(s)}r(s) = \sum_s d_\pi(s)r(s) = \mathbb{E}_{d_\pi}[r(s)].$$

Also,

$$\mathbb{E}_{d_\beta}[w_\pi(s)] = \sum_s d_\beta(s)\frac{d_\pi(s)}{d_\beta(s)} = \sum_s d_\pi(s) = 1.$$

Therefore

$$J_c(\pi) = \mathbb{E}_{d_\beta}[w_\pi r] - \bar{r}_\beta\mathbb{E}_{d_\beta}[w_\pi] = \mathbb{E}_{d_\pi}[r] - \bar{r}_\beta.$$

$\square$

The claim above proves why does performing rewards centering is principled, as well why ZOL on certain benchmarks provide s negligible performance improvements (due to $w_z \approx 1$)

## C.5. Objective: Off-Policy Return with Regularizers

The core (naive) off-policy objective is the reward-weighted importance estimate

$$\widehat{J}(z) \;=\; \frac{1}{B} \sum_{i=1}^{B} w_i(z)\, r_i'.$$

To stabilize optimization, we add two regularizers:

**(i) Chi-square-like ratio variance penalty.** To prevent weight explosion, penalize deviation from 1:

$$\widehat{\chi^2}(z) \;=\; \frac{1}{B} \sum_{i=1}^{B} (w_i(z) - 1)^2.$$

**Theorem C.2** (Chi-square penalty equals a $\chi^2$-divergence). *Let $w_\pi(s) = d_\pi(s)/d_\beta(s)$. Then*

$$\mathbb{E}_{s \sim d_\beta}\left[(w_\pi(s) - 1)^2\right] = \chi^2(d_\pi \,\|\, d_\beta) := \sum_s \frac{(d_\pi(s) - d_\beta(s))^2}{d_\beta(s)}.$$

*Proof.*

$$\mathbb{E}_{d_\beta}[(w_\pi - 1)^2] = \sum_s d_\beta(s) \left(\frac{d_\pi(s)}{d_\beta(s)} - 1\right)^2 = \sum_s d_\beta(s) \frac{(d_\pi(s) - d_\beta(s))^2}{d_\beta(s)^2} = \sum_s \frac{(d_\pi(s) - d_\beta(s))^2}{d_\beta(s)}.$$

$\square$

**(ii) Trust-region penalty in latent space.** Let $\Pi(\cdot)$ denote the model's latent projection/normalization (in code: `project_z`). We penalize deviation from the initial latent:

$$\widehat{\Omega}(z) \;=\; \left\| \Pi(z) - \Pi(z_0) \right\|_2^2.$$

**Final optimization problem.** The algorithm maximizes

$$\max_z \;\; \underbrace{\widehat{J}(z)}_{\text{off-policy reward}} \;-\; \lambda_\chi \underbrace{\widehat{\chi^2}(z)}_{\text{ratio stability}} \;-\; \lambda_z \underbrace{\widehat{\Omega}(z)}_{\text{trust region}} \;,$$

where $\lambda_\chi \geq 0$ and $\lambda_z \geq 0$ are hyperparameters.

## C.6. Gradient-Based Update Rule (Adam)

We optimize $z$ with a first-order optimizer (Adam). At each iteration $t$:

1. Compute $\widehat{\mu}(z_t)$ from cached reset observations.

2. Compute weights $w_i(z_t)$ for batch samples.

3. Compute objective

$$\mathcal{L}(z_t) \;=\; -\widehat{J}(z_t) + \lambda_\chi \widehat{\chi^2}(z_t) + \lambda_z \widehat{\Omega}(z_t).$$

4. Take an Adam step:

$$z_{t+1} \leftarrow \text{AdamUpdate}(z_t, \nabla_z \mathcal{L}(z_t)).$$

5. (Optional) Clip the gradient norm to improve numerical stability:

$$\nabla_z \mathcal{L}(z_t) \leftarrow \nabla_z \mathcal{L}(z_t) \cdot \min\left\{1, \frac{c}{\|\nabla_z \mathcal{L}(z_t)\|_2}\right\}.$$

## C.7. Discussion and Failure Mode Mitigation

This procedure is intentionally simple and purely off-policy: it updates only the latent variable $z$ while keeping all pretrained networks fixed. The stabilization components are essential:

- **Reward centering** prevents optimizing constant reward offsets that favor stationary or *do nothing* behavior.

- **Self-normalized positive weights** reduce variance and prevent trivial scaling of density ratios.

- **Chi-square penalty and clipping** prevent ratio blow-up, a common off-policy pathology where optimization exploits estimator error.

- **Trust-region in latent space** keeps the search near the zero-shot latent and helps avoid catastrophic drift to degenerate skills.

**Proposition C.3** (On-policy (or near-on-policy) data implies no gain from correction). *If $d_\beta = d_\pi$, then $w_\pi(s) \equiv 1$ and the centered objective is identically zero:*

$$J_c(\pi) = \mathbb{E}_{d_\beta}\big[1 \cdot (r - \bar{r}_\beta)\big] = 0.$$

*Therefore any optimization of $J_c(\pi)$ provides no systematic improvement (only regularizers/noise remain).*

*Proof.* If $d_\beta = d_\pi$, then $w_\pi = d_\pi/d_\beta \equiv 1$ and $\mathbb{E}_{d_\beta}[r - \bar{r}_\beta] = 0$. Hence $J_c(\pi) = 0$. □

# D. Experimental Setup

## D.1. Environments

We list the continuous control environments from the DeepMind Control Suite (Tassa et al., 2018) and Humenv (Tirinzoni et al.) used in this work in Table 4.

| Domain | Observation dimension | Action dimension | Episode length |
|---|---|---|---|
| Pointmass | 4 | 2 | 1000 |
| Walker | 24 | 6 | 1000 |
| Cheetah | 17 | 6 | 1000 |
| Quadruped | 78 | 12 | 1000 |
| HumEnv | 358 | 69 | 300 |

*Table 4.* Overview of observation spaces, action spaces and episode length of environments used in this work.

## D.2. ZOL Hyperparameters

**DMC-state.** For state-based DMC environments, the results can be seen in Table 5. Most hyperparameters require almost no additional tuning as well as they are robust (provide marginal decrease/increase in returns).

**OGBench**. For state-based OGBench tasks, we report most suitable ZOL hyperparameters in the Table 6. We note that based on ablation studies and our experiments in the main section of paper, most of the hyperparameters are agnostic to tasks and changing them leads to only marginal increase or decrease in performance. For the FB implementation we additionally used flow-based imitation policy for actor, trained with BC loss.

## D.3. Baselines

**LoLA.** LoLA is an *online latent adaptation* baseline that performs gradient-based search directly in the latent space $z$ of a pretrained behavior foundation model (BFM), rather than updating the full policy network in action space. In our TD3-based LoLA implementation, we sweep the TD3 update-to-data ratio (UTD) in $\{1, 4, 8\}$ and the actor update frequency in $\{1, 4\}$, with actor/critic learning rates fixed to $10^{-4}$. For stability and fast adaptation, we use a small residual network (2-layer MLP, 64 hidden units) when employing a residual critic; otherwise the critic is learned from scratch using a 2-layer MLP with 1024 hidden units. We additionally sweep the warm-start schedule (0 or 5000 steps before updates). For the LoLA-specific search procedure, we sweep the lookahead horizon in $\{50, 100, 250\}$, use 10 total rollouts per update, and use 5 rollouts per

| Domain | Task | $\eta$ | $T$ | $\lambda_\chi$ | $\lambda_z$ | $w_{\max}$ | $N$ |
|---|---|---|---|---|---|---|---|
| Cheetah | run | $5 \times 10^{-4}$ | 200 | 0 | 0.05 | 100 | 256 |
| | run_backward | | 100 | 0.005 | 0.05 | | |
| | walk | | 200 | 0 | 0.05 | | |
| | walk_backward | | 200 | 0.001 | 0.02 | | |
| Pointmass | reach_bottom_left | $5 \times 10^{-4}$ | 200 | 0.005 | 0.02 | 50 | 256 |
| | reach_bottom_right | | | 0.001 | | | |
| | reach_top_left | | | 0.001 | | 100 | |
| | reach_top_right | | | 0.001 | | 50 | |
| Quadruped | jump | $5 \times 10^{-4}$ | 200 | 0 | 0.02 | 100 | 256 |
| | run | | | 0.005 | 0 | | |
| | stand | | | 0 | 0.02 | 50 | |
| | walk | | | 0.005 | 0.05 | 100 | |
| Walker | run | $1 \times 10^{-4}$ | 200 | 0 | 0.05 | 100 | 256 |
| | spin | | 100 | 0.001 | | 50 | |
| | stand | $5 \times 10^{-4}$ | 200 | 0 | 0.02 | 100 | |
| | walk | | | 0.005 | | | |

*Table 5.* **Best ZOL hyperparameters per DMC domain-task**

sampled state to estimate the baseline. We also sweep the reset probability to the initial-state distribution in $\{0, 0.2, 0.5\}$ (otherwise sampling starting states from the replay buffer), and sweep the LoLA step size in $\{0.1, 0.05\}$. Across domains, we train online adaptation methods for 300 episodes with 5 random seeds, and report evaluation returns averaged over 50 episodes.

**ReLA.** is an *online residual latent adaptation* baseline that augments latent-space search with a residual value-learning component to correct for mismatch between the pretrained BFM critic/value estimates and the downstream task. We follow the same TD3-based backbone and hyperparameter sweeps used for the other TD3-based online adaptation baselines (UTD $\in \{1, 4, 8\}$, actor update frequency $\in \{1, 4\}$, actor/critic learning rate $10^{-4}$, warm-start in $\{0, 5000\}$ steps). When using a residual critic, we parameterize the residual with a 2-layer MLP with 64 hidden units; otherwise we learn a critic from scratch with a 2-layer MLP with 1024 hidden units. As with LoLA, we train for 300 online episodes with 5 seeds and evaluate by averaging returns over 50 episodes.

**HILP.** For HILP, we use the official evaluation/training codebase for zero-shot RL and train with TD3 as the underlying offline RL optimizer. For zero-shot RL on DMC/ExORL-style settings, we follow the standard prompting/inference setup where a latent $z$ is either sampled from the prior (probability 0.5) or set to a goal-reaching latent (probability 0.5); at test time, the task-specific latent is inferred via linear regression from a fixed set of transition samples. Hilbert representations are trained with goal relabeling by sampling future states from the same trajectory with probability 0.625 and sampling random states from the dataset with probability 0.375, and we add $\varepsilon = 10^{-6}$ inside the Hilbert-distance computation for numerical stability.

**TD-JEPA.** In all our experiments we take official codebase [2] and take recommended hyperparamters together with neural networks architecture from paper across all benchmarks (ExORL/OGBench).

**FB-CPR.** For the experiments on HumEnv benchmark from §6, we take pretrained FB-CPR from original repository [3]. Thus all hyperparameters of the pretrained model is aligned with original implementation. For the motion tracking, we used averaging over window with horizon 8.

*Networks and architectures.* All networks are MLPs with ReLU activations, except the first hidden layer which uses layer normalization followed by $\tanh$. For $z$-conditioned critics (e.g., $F$ and $Q$) and the actor, we use the embedding-style architecture: two embedding towers (2 layers, 1024 hidden units, output dim 512) followed by a main MLP (2 layers,

---

[2] https://github.com/facebookresearch/td_jepa
[3] https://github.com/facebookresearch/metamotivo

1024 hidden units). Critics are implemented as an ensemble of 2 parallel networks; the actor uses 1 network and outputs a `tanh`-squashed mean with fixed Gaussian standard deviation 0.2. The backward/state embedding $B(s)$ is a simple MLP with one hidden layer of 256 units and $\ell_2$-normalized output (dim 256). The discriminator $D_\psi(s, z)$ is an MLP with 3 hidden layers of 1024 units (first layer: layernorm+tanh, others ReLU) and a sigmoid output; it is trained with learning rate $10^{-5}$ and a WGAN-style gradient penalty with coefficient 10.

*Inference.* For reward-based inference, we compute a latent by weighted regression from the online buffer:

$$z_r \propto \mathbb{E}_{s' \sim D_{\text{online}}} \left[ \exp\left(10\, r(s')\right) B(s')\, r(s') \right],$$

estimating the expectation with 100k samples. For goal-reaching, we use $z_g = B(g)$. For motion tracking of a reference motion $\tau = \{s_j\}$, we infer a time-dependent latent using a sliding window of length $L$ equal to the pre-training sequence length (here $L = 8$):

$$z_t \propto \sum_{j=t+1}^{t+L+1} B(s_j).$$

# E. Code

We provide full training scripts, notebooks, together with visualizations from paper with the submission materials.

# F. Details on Ablation Study

In this section we provide more nuanced discussion on the chosen hyperparameters and how they affect overall return.

### F.1. ExORL DeepMind

**Protocol.** We run a full-factorial sweep over ZOL hyperparameters on the **16 state-based ExORL DeepMind Control tasks** (Walker/Cheetah/Quadruped/PointMass). We report the **mean return over the 16 tasks** (equal weighting). The sweep includes **72 configurations** (**1152** task-level runs; $72 \times 16$). Unless noted, $n_\mu = 256$.

**Overall robustness.** Across ranges, ZOL is stable: for each hyperparameter, the best–worst marginal gap is **¡0.5%** of the mean return.

**Learning dynamics (# optimization steps).** Varying `num_steps` $\in \{100, 200\}$ shows a mild preference for fewer steps: 100 performs best (596.46) vs. 200 (593.82; $-0.44\%$).

**Trust regularization.** For `trust_l2_coef` $\in \{0, 0.02, 0.05\}$, a moderate value is best: 0.02 achieves the highest mean (596.23) vs. 0 (593.99; $-0.38\%$), while 0.05 remains competitive (595.20).

$\chi^2$**-penalty.** With `chi2_coef` $\in \{0, 0.001, 0.005\}$, 0 (595.98) and 0.001 (595.74) are essentially identical, while 0.005 degrades performance (593.71; $-0.38\%$), suggesting large $\chi^2$ penalties can be overly restrictive.

**Weight clipping.** For `weight_clip` $\in \{50, 100\}$, differences are negligible; 100 is slightly better (595.43 vs. 594.85; $+0.10\%$).

### F.2. SMPL Humanoid

We run a full-factorial sweep over ZOL hyperparameters on the **45 SMPL humanoid tasks**. We report the **mean return over the 45 tasks** (equal weighting). The results were obtained using 50 environments and 100 runs per environment. For ablation, we used 10 samples to infer a latent $z$. The sweep includes **15 configurations** (**675** task-level runs; $15 \times 45$).

**Overall robustness.** Across ranges, ZOL is stable: for each hyperparameter, the best–worst marginal gap is **¡0.5%** of the mean return.

**Learning dynamics (# optimization steps).** Varying `num_steps` $\in \{100, 150, 200\}$ shows a mild preference for moderate steps: 150 performs best (151.38) vs. 100 (150.96; $-0.28\%$) and 200 (151.26).

**Trust regularization.** For `trust_l2_coef` $\in \{0.01, 0.1, 0.5\}$, results are very stable with a slight preference for higher regularization: 0.5 achieves the highest mean (151.29) vs. 0.1 (151.15; $-0.09\%$), while 0.01 remains competitive (151.23).

$\chi^2$**-penalty.** With chi2_coef $\in \{0.01, 0.1, 0.5\}$, smaller penalties are preferred: 0.01 performs best (151.50), while 0.5 degrades performance slightly (151.03; $-0.31\%$), suggesting large $\chi^2$ penalties can be overly restrictive.

**Weight clipping.** For weight_clip $\in \{50, 100\}$, differences are negligible; 50 is nominally better (151.20 vs. 151.20; $< 0.01\%$).

## G. Discussion and potential impact

### G.1. Behavior-Supported Policy Parametrization

To enforce that the target policy $\pi$ remains within the support of the dataset, one would like to guarantee that

$$\pi(a \mid s) = 0 \quad \text{whenever} \quad d^\beta(s, a) = 0. \tag{13}$$

The low-rank density ratio estimator (10) define a *behavior-supported* policy by reweighting the behavior policy $\beta$ with the estimated ratio:

$$\pi_z(a \mid s) \propto w_{\pi_z}(s, a) \, \beta(a|s) \approx (1 - \gamma) \, W_{\pi_z}^\top B(s, a) \, \beta(a|s), \tag{14}$$

with normalization $\sum_a \pi(a \mid s){=}1$ for all $s$. This parametrization has several immediate consequences. First, it enforces the support constraint, because $\beta(a \mid s) = 0$ implies $\pi(a \mid s) = 0$ by construction. Thus $\pi(\cdot \mid s)$ is absolutely continuous with respect to $\beta(\cdot \mid s)$, and the learned policy never assigns positive probability to unseen actions. Second, the policy inherits the low-rank structure of the FB representation: it depends on $(s, a)$ only through the backward embedding $B(s, a)$ and on the policy $\pi$ only through the forward expectation vector $W\pi$. This acts as a structural regularizer on the policy class. Third, the identity

$$\mathbb{E}_{d^\pi}[f] = \mathbb{E}_{d^\beta}[w_\pi f]$$

continues to hold by construction, so that ZOL remains compatible with the stationary distribution correction frameworks used in DualDICE and OptiDICE. In this sense, the FB representation provides a concrete parametrization of $w_\pi$ that fits naturally into existing DICE-style algorithms.

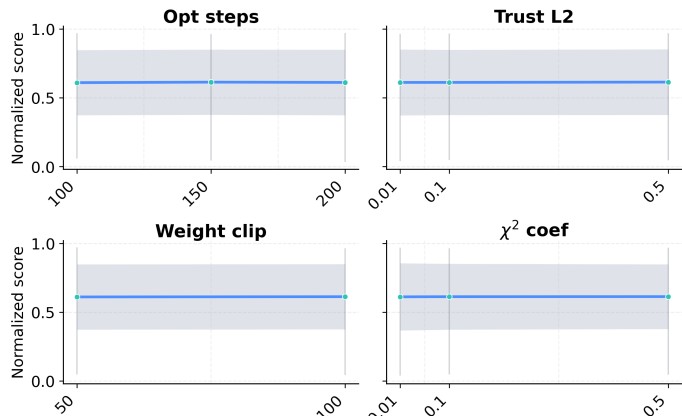

*Figure 8.* Hyperparameters ablation on the normalized return on the SMPL Humanoid benchmark.

| Domain | Task | $\eta$ | $T$ | $\lambda_\chi$ | $\lambda_z$ | $w_{\max}$ | $N$ |
|---|---|---|---|---|---|---|---|
| antmaze-large-navigate-v0 | antmaze-large-navigate-singletask-task1-v0 | $1 \times 10^{-3}$ | 200 | 0.05 | 0.005 | 20 | 512 |
| | antmaze-large-navigate-singletask-task2-v0 | $3 \times 10^{-3}$ | 200 | 0.05 | 0.005 | 50 | 512 |
| | antmaze-large-navigate-singletask-task3-v0 | $3 \times 10^{-3}$ | 500 | 0.01 | 0.001 | 50 | 512 |
| | antmaze-large-navigate-singletask-task4-v0 | $3 \times 10^{-3}$ | 200 | 0.05 | 0.005 | 20 | 512 |
| | antmaze-large-navigate-singletask-task5-v0 | $1 \times 10^{-3}$ | 200 | 0.05 | 0.005 | 50 | 512 |
| antmaze-large-stitch-v0 | antmaze-large-stitch-singletask-task1-v0 | $1 \times 10^{-3}$ | 200 | 0.01 | 0.001 | 50 | 1 |
| | antmaze-large-stitch-singletask-task2-v0 | $1 \times 10^{-3}$ | 200 | 0.01 | 0.001 | 50 | 1 |
| | antmaze-large-stitch-singletask-task3-v0 | $1 \times 10^{-3}$ | 200 | 0.01 | 0.001 | 50 | 512 |
| | antmaze-large-stitch-singletask-task4-v0 | $1 \times 10^{-3}$ | 200 | 0.01 | 0.001 | 50 | 512 |
| | antmaze-large-stitch-singletask-task5-v0 | $1 \times 10^{-3}$ | 200 | 0.01 | 0.001 | 50 | 512 |
| antmaze-medium-explore-v0 | antmaze-medium-explore-singletask-task1-v0 | $1 \times 10^{-3}$ | 200 | 0.01 | 0.001 | 50 | 512 |
| | antmaze-medium-explore-singletask-task2-v0 | $1 \times 10^{-3}$ | 200 | 0.01 | 0.001 | 50 | 1 |
| | antmaze-medium-explore-singletask-task3-v0 | $1 \times 10^{-3}$ | 200 | 0.01 | 0.001 | 50 | 512 |
| | antmaze-medium-explore-singletask-task4-v0 | $1 \times 10^{-3}$ | 200 | 0.01 | 0.001 | 50 | 512 |
| | antmaze-medium-explore-singletask-task5-v0 | $1 \times 10^{-3}$ | 200 | 0.01 | 0.001 | 50 | 512 |
| antmaze-medium-navigate-v0 | antmaze-medium-navigate-singletask-task1-v0 | $1 \times 10^{-3}$ | 200 | 0.05 | 0.001 | 50 | 512 |
| | antmaze-medium-navigate-singletask-task2-v0 | $1 \times 10^{-3}$ | 200 | 0.05 | 0.001 | 50 | 512 |
| | antmaze-medium-navigate-singletask-task3-v0 | $1 \times 10^{-3}$ | 500 | 0.05 | 0.005 | 20 | 512 |
| | antmaze-medium-navigate-singletask-task4-v0 | $3 \times 10^{-3}$ | 200 | 0.01 | 0.005 | 20 | 512 |
| | antmaze-medium-navigate-singletask-task5-v0 | $1 \times 10^{-3}$ | 200 | 0.01 | 0.005 | 50 | 512 |
| antmaze-medium-stitch-v0 | antmaze-medium-stitch-singletask-task1-v0 | $1 \times 10^{-3}$ | 500 | 0.05 | 0.005 | 20 | 512 |
| | antmaze-medium-stitch-singletask-task2-v0 | $3 \times 10^{-3}$ | 500 | 0.05 | 0.005 | 20 | 512 |
| | antmaze-medium-stitch-singletask-task3-v0 | $5 \times 10^{-3}$ | 500 | 0.01 | 0.001 | 50 | 512 |
| | antmaze-medium-stitch-singletask-task4-v0 | $3 \times 10^{-3}$ | 200 | 0.01 | 0.001 | 50 | 512 |
| | antmaze-medium-stitch-singletask-task5-v0 | $1 \times 10^{-3}$ | 200 | 0.05 | 0.001 | 50 | 512 |
| cube-double-play-v0 | cube-double-play-singletask-task1-v0 | $1 \times 10^{-3}$ | 200 | 0.01 | 0.001 | 50 | 1 |
| | cube-double-play-singletask-task2-v0 | $1 \times 10^{-3}$ | 200 | 0.01 | 0.001 | 50 | 1 |
| | cube-double-play-singletask-task3-v0 | $1 \times 10^{-3}$ | 200 | 0.01 | 0.001 | 50 | 1 |
| | cube-double-play-singletask-task4-v0 | $1 \times 10^{-3}$ | 200 | 0.01 | 0.001 | 50 | 1 |
| | cube-double-play-singletask-task5-v0 | $1 \times 10^{-3}$ | 200 | 0.01 | 0.001 | 50 | 1 |
| cube-single-play-v0 | cube-single-play-singletask-task1-v0 | $1 \times 10^{-3}$ | 200 | 0.01 | 0.001 | 50 | 1 |
| | cube-single-play-singletask-task2-v0 | $1 \times 10^{-3}$ | 200 | 0.01 | 0.001 | 50 | 512 |
| | cube-single-play-singletask-task3-v0 | $1 \times 10^{-3}$ | 200 | 0.01 | 0.001 | 50 | 1 |
| | cube-single-play-singletask-task4-v0 | $1 \times 10^{-3}$ | 200 | 0.01 | 0.001 | 50 | 512 |
| | cube-single-play-singletask-task5-v0 | $1 \times 10^{-3}$ | 200 | 0.01 | 0.001 | 50 | 512 |
| puzzle-3x3-play-v0 | puzzle-3x3-play-singletask-task1-v0 | $1 \times 10^{-3}$ | 200 | 0.01 | 0.001 | 50 | 512 |
| | puzzle-3x3-play-singletask-task2-v0 | $1 \times 10^{-3}$ | 200 | 0.01 | 0.001 | 50 | 1 |
| | puzzle-3x3-play-singletask-task3-v0 | $1 \times 10^{-3}$ | 200 | 0.01 | 0.001 | 50 | 1 |
| | puzzle-3x3-play-singletask-task4-v0 | $1 \times 10^{-3}$ | 200 | 0.01 | 0.001 | 50 | 1 |
| | puzzle-3x3-play-singletask-task5-v0 | $1 \times 10^{-3}$ | 200 | 0.01 | 0.001 | 50 | 1 |

*Table 6.* **Best ZOL hyperparameters per OGBench domain-task.**

*Table 7.* **FB vs. ZOL comparison on all tasks.** We report mean $\pm$ std over evaluation rollouts. ZOL (Ours) is highlighted, along with the absolute improvement $\Delta$ and relative gain where applicable.

| TASK | FB | ZOL (OURS) | $\Delta$(IMPROVEMENT) |
|---|---|---|---|
| CRAWL-0.4-0-D | 173.92 $\pm52$ | **201.77** $\pm39$ | **+27.85** (16.01%) |
| CRAWL-0.4-0-U | 108.77 $\pm6$ | **110.53** $\pm5$ | **+1.76** (1.62%) |
| CRAWL-0.4-2-D | **18.73** $\pm7$ | 9.36 $\pm4$ | -9.37 |
| CRAWL-0.4-2-U | 13.60 $\pm3$ | **20.55** $\pm6$ | **+6.95** (51.10%) |
| CRAWL-0.5-0-D | 167.79 $\pm42$ | **192.44** $\pm41$ | **+24.65** (14.69%) |
| CRAWL-0.5-0-U | 97.75 $\pm6$ | **110.39** $\pm5$ | **+12.64** (12.93%) |
| CRAWL-0.5-2-D | **34.68** $\pm8$ | 26.96 $\pm9$ | -7.72 |
| CRAWL-0.5-2-U | 18.07 $\pm3$ | **20.80** $\pm5$ | **+2.73** (15.11%) |
| CROUCH-0 | 243.90 $\pm16$ | **249.98** $\pm8$ | **+6.08** (2.49%) |
| HEADSTAND | 30.93 $\pm22$ | **40.26** $\pm29$ | **+9.33** (30.16%) |
| JUMP-2 | **38.80** $\pm1$ | 38.19 $\pm1$ | -0.61 |
| LIEONGROUND-DOWN | 196.89 $\pm13$ | **204.78** $\pm23$ | **+7.89** (4.01%) |
| LIEONGROUND-UP | 216.69 $\pm3$ | **219.38** $\pm9$ | **+2.69** (1.24%) |
| MOVE-EGO–90-2 | 213.51 $\pm3$ | **234.58** $\pm2$ | **+21.07** (9.87%) |
| MOVE-EGO–90-4 | **215.37** $\pm3$ | 207.64 $\pm3$ | -7.73 |
| MOVE-EGO-0-0 | 269.19 $\pm4$ | **271.14** $\pm5$ | **+1.95** (0.72%) |
| MOVE-EGO-0-2 | 261.33 $\pm2$ | **264.99** $\pm2$ | **+3.66** (1.40%) |
| MOVE-EGO-0-4 | 253.32 $\pm4$ | **254.29** $\pm2$ | **+0.97** (0.38%) |
| MOVE-EGO-180-2 | 219.57 $\pm2$ | **232.52** $\pm2$ | **+12.95** (5.90%) |
| MOVE-EGO-180-4 | 205.80 $\pm2$ | **205.82** $\pm1$ | **+0.02** (0.01%) |
| MOVE-EGO-90-2 | 228.33 $\pm3$ | **240.53** $\pm3$ | **+12.20** (5.34%) |
| MOVE-EGO-90-4 | 183.23 $\pm2$ | **186.17** $\pm2$ | **+2.94** (1.60%) |
| MOVE-EGO-LOW–90-2 | **248.29** $\pm4$ | 247.75 $\pm5$ | -0.54 |
| MOVE-EGO-LOW-0-0 | 198.20 $\pm23$ | **200.52** $\pm23$ | **+2.32** (1.17%) |
| MOVE-EGO-LOW-0-2 | 214.44 $\pm3$ | **252.94** $\pm4$ | **+38.50** (17.95%) |
| MOVE-EGO-LOW-180-2 | 114.21 $\pm5$ | **119.47** $\pm3$ | **+5.26** (4.61%) |
| MOVE-EGO-LOW-90-2 | 235.99 $\pm3$ | **239.78** $\pm6$ | **+3.79** (1.61%) |
| RAISEARMS-H-H | 248.58 $\pm6$ | **249.39** $\pm5$ | **+0.81** (0.33%) |
| RAISEARMS-H-L | 42.55 $\pm1$ | **241.40** $\pm4$ | **+198.85** (467.33%) |
| RAISEARMS-H-M | **243.15** $\pm10$ | 183.94 $\pm20$ | -59.21 |
| RAISEARMS-L-H | 188.61 $\pm5$ | **256.10** $\pm3$ | **+67.49** (35.78%) |
| RAISEARMS-L-L | 265.32 $\pm3$ | **268.07** $\pm4$ | **+2.75** (1.04%) |
| RAISEARMS-L-M | **257.94** $\pm4$ | 160.34 $\pm8$ | -97.60 |
| RAISEARMS-M-H | **224.76** $\pm12$ | 175.52 $\pm14$ | -49.24 |
| RAISEARMS-M-L | 85.16 $\pm4$ | **133.25** $\pm9$ | **+48.09** (56.47%) |
| RAISEARMS-M-M | **258.12** $\pm4$ | 222.33 $\pm16$ | -35.79 |
| ROTATE-X–5-0.8 | 3.18 $\pm2$ | **17.89** $\pm3$ | **+14.71** (462.58%) |
| ROTATE-X-5-0.8 | 2.02 $\pm2$ | **18.83** $\pm2$ | **+16.81** (832.18%) |
| ROTATE-Y–5-0.8 | 159.39 $\pm7$ | **225.83** $\pm8$ | **+66.44** (41.68%) |
| ROTATE-Y-5-0.8 | **134.09** $\pm5$ | 124.09 $\pm5$ | -10.00 |
| ROTATE-Z–5-0.8 | **102.32** $\pm21$ | 81.32 $\pm9$ | -21.00 |
| ROTATE-Z-5-0.8 | 136.08 $\pm3$ | **137.82** $\pm8$ | **+1.74** (1.28%) |
| SITONGROUND | **215.17** $\pm6$ | 211.94 $\pm9$ | -3.23 |
| SPLIT-0.5 | **252.20** $\pm3$ | 250.60 $\pm6$ | -1.60 |
| SPLIT-1 | **131.52** $\pm53$ | 53.46 $\pm23$ | -78.06 |

