# OpenReview forum: "Zero-Shot Off-Policy Learning"
_ICML.cc/2026/Conference — ICML 2026 regular_

### Official Review · Reviewer_fvTf · 2026-02-18

**Soundness:** 4
**Presentation:** 4
**Significance:** 4
**Originality:** 3
**Overall Recommendation:** 5
**Confidence:** 3

**Summary:**

The paper proposes a zero-shot off-policy learning method that can be used for distribution correction without any interactions to the environment. Based on the theoretical discovery on the connection between successor measure and stationary density ratios, the paper formulates the zero-shot adaptation method by inferring the optimal importance sampling ratio. Empirical results on ExORL, OGBench and SMPL Humanoid benchmarks show convincing improvements over existing baselines.

**Compliance With Llm Reviewing Policy:**

Affirmed.

**Final Justification:**

The rebuttal addressed my main concern.

**Key Questions For Authors:**

See the weakness section.

**Limitations:**

yes

**Strengths And Weaknesses:**

## Strength

- The paper introduces a novel method on zero-shot off-policy learning. The overall presentation of the paper is clear and the statements are easy to follow.

- The paper is based on a theoretical connection between density ratios and successor measure, which is theoretically sound.

- The paper provides extensive empirical validation to their proposed method over diverse domains, demonstrating the effectiveness of their proposed method. The paper also provides a toy example on the failure case comparing their approach to FB, which is quite intuitive.

## Weaknesses

- It would strengthen the paper more if the author could elaborate on the improvement over FB by providing more in-depth theoretical analysis on the failure case.

---

> ### Author Rebuttal · Authors · 2026-03-29
>
> We thank the reviewer for the positive assessment and for this helpful suggestion. We agree that paper claims can be strengthened with a more explicit theoretical characterization of the FB failure mode and how ZOL improves upon it. We provide an additional, simple experiment below.
>
> Consider the state-only setting of Appendix C. There are three states: an initial state $s_0$ and two absorbing states $A,B$. From $s_0$, action $a_A$ transitions deterministically to $A$, and action $a_B$ transitions deterministically to $B$. The behavior policy satisfies
> $$
> \beta(a_A\mid s_0)=p,\qquad \beta(a_B\mid s_0)=1-p
> $$
> with $p\in(1/2,1)$. Let the rewards be
> $$
> r(s_0)=0,\qquad r(A)=r_A,\qquad r(B)=r_B
> $$
> with $r_B>r_A>0$
>
> Now choose an exact reward basis
> $$
> B(s_0)=0,\qquad B(A)=e_1,\qquad B(B)=e_2,
> $$
> and exact successor-feature scores at the decision state
> $$
> F(s_0,a_A,z)^\top z = cz_1,\qquad F(s_0,a_B,z)^\top z = cz_2
> $$
> for some constant $c>0$. Then the greedy FB policy is
> $$
> \pi_z(s_0)=a_A \iff z_1>z_2,
> $$
> so the representation is expressive enough to distinguish the two behaviors exactly.
>
> Under the discounted behavior occupancy,
> $$
> d^\beta(s_0)=1-\gamma,\qquad d^\beta(A)=\gamma p,\qquad d^\beta(B)=\gamma(1-p).
> $$
> Hence the vanilla FB latent from Eq. (8) is
> $$
> z_{\mathrm{FB}}
> = E_{s\sim d^\beta}[B(s)r(s)]
> = \gamma p r_A e_1 + \gamma(1-p) r_B e_2.
> $$
> Therefore FB chooses action $a_A$ whenever
> $$
> pr_A > (1-p)r_B
> $$
> However, the normalized on-policy returns are
> $$
> E_{d^{\pi_A}}[r]=\gamma r_A,\qquad
> E_{d^{\pi_B}}[r]=\gamma r_B,
> $$
> so the optimal policy is $\pi_B$ whenever $r_B>r_A$.
>
> Thus, whenever
> $$
> r_B>r_A
> \qquad\text{but}\qquad
> pr_A>(1-p)r_B,
> $$
> vanilla FB provably selects the wrong latent/policy despite an exact reward basis and exact greedy extractor. A concrete example is $p=0.9$, $r_A=1$, $r_B=4$: FB chooses $A$ although $B$ is strictly better.
>
> ZOL fixes exactly this inference-time failure. For the two candidate policies, the exact density ratios are
> $$
> w_A(s_0)=1,\quad w_A(A)=1/p,\quad w_A(B)=0,
> $$
> $$
> w_B(s_0)=1,\quad w_B(A)=0,\quad w_B(B)=1/(1-p)
> $$
> By Theorem in Appendix C.1, the centered objective satisfies
> $$
> J_c(\pi)
> := E_{d^\beta}\left[w_{\pi/\beta}(s)(r(s)-\bar r_\beta)\right]
> = E_{d^\pi}[r]-\bar r_\beta
> $$
> Therefore
> $$
> J_c(\pi_B)-J_c(\pi_A)=\gamma(r_B-r_A)>0
> $$
> independently of the dataset bias $p$.
>
> So, in this exact toy MDP, FB fails because Eq 8 performs latent selection under the behavior occupancy $d^\beta$, which overweights high-support regions/actions. ZOL instead evaluates each candidate latent under its induced occupancy via $w_{\pi/\beta}$, and therefore recovers the correct policy ranking. Moreover, the argument works without centering of rewards, since exact reweighting satisfies $E_{d^\beta}[w_\pi r] = E_{d^\pi}[r]$.
>
> We will add this analysis in the revision to complement the donut visualization in Fig. 5.

---

> > ### Author Rebuttal · Reviewer_fvTf · 2026-04-02
> >
> > My concerns have been adequately addressed.

---

### Official Review · Reviewer_wAvq · 2026-03-06

**Soundness:** 3
**Presentation:** 2
**Significance:** 3
**Originality:** 3
**Overall Recommendation:** 4
**Confidence:** 3

**Summary:**

This paper studies a practical problem in offline reinforcement learning: how to adapt a policy to a new task without retraining from scratch. The key difficulty is that, at test time, the latent selected for the new task may induce policies that visit state-action regions poorly covered by the offline dataset, which makes value estimation unreliable and hurts performance. The paper argues that, in many zero-shot methods, the main issue is not necessarily the representation itself, but the test-time latent selection step.

The method builds on the Forward-Backward (FB) framework (2021), where representations are learned from reward-free offline data and then used to recover a task-specific policy once a new reward is given. The main contribution of this paper is to introduce an off-policy distribution correction into zero-shot latent inference. More specifically, the authors derive a ratio-based correction term and use it to guide latent optimization toward policies that are both relevant to the new task and better supported by the dataset. The idea is interesting, and the paper presents experiments on multiple benchmarks to support the approach.

**Compliance With Llm Reviewing Policy:**

Affirmed.

**Ethical Review Concerns:**

The core algorithm in the main paper is not consistent with the authors' rebuttal. This conflict

**Final Justification:**

Thanks to the authors for the clarifications on the method. The algorithm sounds fair to me, but the authors should be careful about using "near-optimal" in their claim, which is not strictly proved in this paper. I will raise the score.

**Key Questions For Authors:**

Questions are mostly as I mentioned in weakness:
1. Mismatch claim
2. Can the authors provide a more formal error propagation discussion for Eq. (10)? For example, how do FB approximation error, density-ratio clipping, and trust-region constraints affect final performance?
3. source of improvement

A few more questions:
1. To what extent does the method depend on the quality of FB factorization? If FB is learned imperfectly, does the ratio estimation become unreliable?
2. What is the computational cost in practice? How many steps does test-time latent optimization require, what is the per-step cost, and does it rely on multiple rollouts or simulator calls?

**Limitations:**

Yes

**Strengths And Weaknesses:**

Strengths

This paper addresses a meaningful and relevant problem. Zero-shot adaptation in offline RL is attractive in principle, but it is often fragile because the inferred test-time policy may go outside dataset support. The paper identifies this issue clearly and provides a reasonable motivation for addressing it.

The connection between FB representations and occupancy / density-ratio correction is interesting and, in my view, the most valuable part of the paper. It gives a more principled perspective on why zero-shot latent inference may fail and how one might correct it.

The method is also reasonably natural as an extension of the FB framework. It does not require changing the entire training pipeline, and the empirical evaluation covers several benchmarks, which helps demonstrate that the idea is not limited to a single setting.

Weaknesses

My main concern is that the paper’s claims around “zero-shot,” “interaction-free,” and “fully offline” seem too stronger than the actual algorithm. From the algorithm description, estimating the proposed correction needs to require additional samples or distribution estimates under the new task. If test-time interaction or rollout is needed to train the weights, then the method is no longer strictly interaction-free in the usual sense. The paper should clarify more explicitly how test-time environment interaction is required, how such data is obtained, and whether this setting should still be called strict zero-shot / interaction-free adaptation.

A second concern is that the “near-optimal” claim is not sufficiently supported. The theory is interesting, but the final method relies on several approximations, including imperfect FB factorization, approximate reward decomposition, approximate ratio estimation, and multiple stabilization components such as clipping and trust-region constraints. Under these approximations, the current "near-optimal" seems more like a useful intuition than a strong guarantee.

Third, the experiments do not always clearly support the conclusion that the method is overall better than FB. Some numbers in the tables appear worth double-checking. For example, in Table 1, if higher is better, some entries do not clearly favor ZOL over FB as claimed. The averaging and rounding also seem worth verifying (from my rough calculation, the avg of FB should be 0.4569, which might not be rounded to 0.45 because the proposed ZOL is 0.47. In addition, if we remove the third type PUZZLE-3X3-PLAY tasks, FB is even better than ZOL. The numerical reporting in the tables should be checked carefully, since even small inconsistencies in averaging or comparison can affect how convincing the results appear.

Finally, it is still unclear how much of the improvement comes from the core correction term itself and how much comes from the additional regularization and stabilization techniques. A more careful ablation would help isolate the main source of performance gains.

---

> ### Author Rebuttal · Authors · 2026-03-28
>
> We thank the reviewer for the careful reading and constructive feedback. We address each point below
>
> **Q1 - Zero-shot / interaction-free claim**
>
> We believe this point stems from a misunderstanding of how the correction term is computed at inference time. ZOL is requires no environment rollouts at test time, no collection of new transitions, and no new reward observations. More specifically, for a candidate latent $z$, the expected forward embedding
> $$W_{\pi_z} = \mathbb{E}_{s_0 \sim \rho_0,\, a_0 \sim \pi_z(\cdot \mid s_0)}\left[F(s_0,a_0,z)\right]$$
> estimated as
>
> $$\hat W_{\pi_z}=\frac{1}{N}\sum_{j=1}^N F_\theta(s_0^{(j)},\pi_z(s_0^{(j)}),z)$$
> Monte Carlo over a batch of initial states $s_0$. So, that density ratio is computed by forward passes through the pretrained FB embeddings on initial states. Crucially, **these initial states are not obtained by additionally interacting with the environment at test time**. We also refer the reviewer to our answer to the question Q3 of the reviewer D6HX. We will revise the paper to make this point fully clear
>
> ---
>
> **Q2 - Optimality Support. The current "near-optimal" seems more like a useful intuition than a strong guarantee**
>
> We agree that the current phrasing is too strong. Our theory motivates the objective in the idealized setting, but the practical algorithm relies on approximate FB factorization, approximate reward decomposition, approximate ratio estimation, and several stabilization components. We will therefore soften the wording from “near-optimal” to a more accurate claim such as “improved task-aligned and dataset-consistent adaptation”, which better reflects what is supported by the paper.
>
> ---
>
> **Q3-Empirical Consistency. The averaging and rounding also seem worth verifying ... if we remove the third type ```puzzle-3x3-play``` tasks, FB is even better than ZOL.**
>
> Summing the exact reported FB scores across the 8 tasks yields: $0.68+0.36+0.69+0.62+0.95+0.0+0.31+0.0 = 3.61$. Dividing by 8 tasks gives a true mean of $0.45125$, which correctly rounds to $0.45$. Summing the ZOL scores gives $3.73$, which yields a mean of $0.46625$, rounding to $0.47$
>
> Excluding tasks (```puzzle-3x3```) where the baseline completely fails ($0.0$) changes the average. However, "cherry-picking" by removing the hardest tasks defeats the purpose of evaluating a generalist foundation model. A core strength of ZOL is its ability to unlock capabilities in environments where vanilla inference completely collapses.
>
> ---
>
> **Q4-Ablation Clarity. It is still unclear how much of the improvement comes from the core correction term itself and how much comes from the additional regularization...**
>
> The data directly answers this question in Appendix F. The vast majority of the improvement comes from the core correction term. For instance, on the ExORL suite, the baseline vanilla FB achieves an average return of $592.5$. When we run ZOL with the trust region penalty turned completely off (trust_l2_coef=0), the method still achieves $593.99$, and the optimal regularizer (0.02) only pushes it to $596.23$. Similarly, turning off the $\chi^2$ penalty yields $595.98$, practically identical to the best setting of $595.74$. This definitively proves that the core density-ratio objective is the primary driver of performance, while the regularizers merely provide marginal stability against edge-case divergence.
>
> ---
>
> **Q5-Error Propagation & Factorization Dependence. How do FB approximation error... affect final performance? To what extent does the method depend on the quality of FB factorization?**
>
> We agree that ZOL depends on the quality of the pretrained FB. Eq. (10) uses the low-rank estimator
> $$
> \hat w_z(s)=(1-\gamma)\widehat{\mu}(z)^\top B^\theta(s)
> $$
> with $$\hat \mu(z):=E_{(s_0,a_0)\sim \rho_0}[F_\theta(s_0,a_0,z)]$$ and let the optimal quantities be $\mu^\* (z), B^\*(s)$, with
>
> $$w_z^\*(s)=(1-\gamma){\mu^\*(z)}^\top B^\*(s)$$
>
> Then, for any in-support $s$,
> $$\|\hat w_z(s)-w_z^\*(s)\|\le(1-\gamma)\left(
> ||\hat\mu(z)-\mu^\*(z)||_2 ||\text{B}^\theta (s)||_2 +
> ||\mu^\*(z)||_2 ||\text{B}^\theta - \text{B}^\*(s)||_2\right)$$
>
> Thus, the ratio-estimation error grows linearly with forward/backward approximation error.
>
> Our centered return objective is
> $$
> \hat J(z)=\frac{1}{B}\sum_{i=1}^B w_i(z)r_i',
> $$
> where
> $$
> r_i' = r_i - \frac{1}{B}\sum_{j=1}^B r_j.
> $$
> If $|r_i'| \le R_{\max}$, then
> $$
> |\hat J(z)-J^\star(z)|
> \le
> R_{\max}\frac{1}{B}\sum_{i=1}^B |w_i(z)-w_i^\star(z)|.
> $$
> Hence, imperfect FB factorization directly perturbs the weighted-return objective. This is why we state in the limitations that ZOL inherits the representational bias of the learned FB factorization
>
> ---
> **Q6-Computational Cost. Test-time optimization steps and rollout requirement**
>
> Test-time cost is low. **ZOL optimizes only the low-dimensional latent $z$ without any parameter updates to model using given offline dataset**. There are no online rollouts. $100–200$ steps are sufficient for convergence across all environments

---

> > ### Author Rebuttal · Reviewer_wAvq · 2026-04-03
> >
> > I understand that the method does not require online rollouts during deployment. However, my concern is whether it still requires interaction with the new task or target environment before deployment/adaptation. From my reading, the method still seems to rely on such interaction, even if not during the final deployment itself. If so, this appears inconsistent with the paper’s central claims of being “zero-shot,” “interaction-free,” and “fully offline.”

---

> > > ### Author Response · Authors · 2026-04-03
> > >
> > > Dear Reviewer, **our method does not require any interactions with the new task or target environment, neither before adaptation nor during adaptation itself**.  Our pipeline is offline and relies strictly on the original pretraining dataset (we denote its occupancy as $d^\beta$ in paper). The process works as follows:
> > >
> > > 1) **Interaction-Free Pretraining**: We pretrain our Behavioral Foundation Model (BFM) using Forward-Backward (FB) method in an unsupervised manner using a static, reward-free offline dataset (*Standard FB step*) [1,2,3].
> > >
> > > 2) **Interaction-Free Latent $z$ Inference**: To adapt to a new problem, we do not perform rollouts. Instead, given a target reward function, we simply label each state in the pre-training dataset with its corresponding reward according to that function (Standard FB step). We infer the latent vector $z$, which conditions the BFM to produce the target policy, using only this preexisting offline data , Eq. 8. (*Standard FB step*)[1,2,3]
> > >
> > > 3) **Interaction-Free $z$ Improvement**: We refine the vector $z$ to mitigate overestimation bias and distribution shift using our theoretical framework and the ZOL algorithm. This step is conducted entirely on the static offline dataset, without a single environment step (*Our method*).
> > >
> > > **By design, environment interaction is completely absent from our method**. We follow the established zero-shot RL pipelines found in prior literature [1, 2, 3]. Our method is based only on the given offline dataset, this is explained in our Introduction (lines 103-107) in our Method section (Section 3, lines 201–204) and reiterated explicitly in Related Work (Section 4, lines 263–268). We would greatly appreciate it if reviewer could point out which specific part of our paper gave the impression that we assume environment interactions.
> > >
> > > We appreciate the reviewer's comments. So, we thank again for your careful attention to detail and we are happy to revise the text accordingly to ensure that this strictly zero-shot property is unmistakable.
> > >
> > >  - Learning One Representation to Optimize All Rewards (https://arxiv.org/pdf/2103.07945)
> > >
> > >  - Zero-Shot Whole-Body Humanoid Control via Behavioral Foundation Models (https://arxiv.org/pdf/2504.11054)
> > >
> > >  - BFM-Zero: A Promptable Behavioral Foundation Model for Humanoid Control Using Unsupervised Reinforcement Learning (https://arxiv.org/pdf/2511.04131)

---

### Official Review · Reviewer_vwwb · 2026-03-11

**Soundness:** 3
**Presentation:** 2
**Significance:** 3
**Originality:** 3
**Overall Recommendation:** 4
**Confidence:** 3

**Summary:**

This manuscript focuses on the issue of distribution shift and value function overestimation in zero-shot off-policy RL. The authors propose Zero-Shot Off-policy Learning (ZOL) algorithm, and provide rigorous derivation on Successor Measures (SMs) and stationary density ratios. The proposed method is evaluated on several benchmark tasks, and experimental results show that ZOL achieves consistent improvements in robustness and performance across all benchmarks.

**Compliance With Llm Reviewing Policy:**

Affirmed.

**Final Justification:**

The authors' rebuttal has addressed my main concerns

**Key Questions For Authors:**

* The manuscript is somehow difficult to follow. When introducing the problem, could the authors provide a more concrete example?  For Section 3, the reviewer suggests to put more useful

* One issue in ZOL is that the performance of ZOL depends on the quality of FB representation. If the low-rank decomposition of FB has biases in density ratio estimation, the ZOL method could hardly identify and correct it.

* The authors introduce a $\chi^2$ penalty in this manuscript for penalizes high variance. It is actually not common to see this regularization term, and could the authors do some comparison on this regularization and other ones?

* The superiority of ZOL is verified on motion tracking tasks , continuous control and long-horizon tasks. How is the performance on discrete RL tasks or real-world robot tasks? Could the authors share some results?


* From the results in table 1, although the performance of ZOL is satisfying, the results of ZOL is quite close to other methods for Antmaze task, any explanations on this issue?

* One typo to point out, in Antmaze-L-Stitch task, FB serves as the best solution?

**Limitations:**

See the questions above

**Strengths And Weaknesses:**

* The novelty and originality of the manuscript is good. The proposed ZOL framework not only combine the merits of existing methods, but also provide an end-to-end solution based on the theoretical discovery

* The idea of ZOL is appealing. In ZOL, only inference optimization is performed at test time without changing the pretraining process of BFMs; it is fully based on offline data and requires no online environment interaction.

*  The manuscript explicitly addresses distribution shift via stationary occupancy correction, and avoids policy extrapolation to out-of-pretraining-data regions, effectively suppressing value function overestimation bias, which is practical solution for offline RL problems.

*  The proposed method has strong compatibility with existing methods. It can be seamlessly integrated into existing FB representation frameworks without major modifications to current BFMs, featuring good engineering implementability.

---

> ### Author Rebuttal · Authors · 2026-03-28
>
> We thank the reviewer for the positive assessment of the novelty, soundness, and practical relevance of our work, and for the constructive suggestions on clarity, limitations, and empirical scope. We address each point below.
>
> **Q1 - Conceptual Clarity**
>
> We appreciate this suggestion and agree that the paper would benefit from a more concrete entry point. To this end, we will revise the introduction to include an earlier intuitive explanation of the failure mode we target: standard zero-shot FB inference can select a latent that is reward-aligned but induces occupancy outside the support of the offline dataset, which can lead to unreliable value estimates and over-optimistic policy selection.
>
> We will also add an explicit forward reference to Section 5, where we already provide a 2D toy example designed to isolate and visualize this issue. Our intention in Section 3 was to formally establish the link between FB representations and DICE-style occupancy correction, but we agree that a clearer intuition should appear earlier.
>
> ---
>
> **Q2 - Representation Bias**
>
> We agree with the reviewer. ZOL depends on the quality of the pretrained FB representation and cannot fully correct errors caused by a poor low-rank factorization. We explicitly acknowledge this in the limitation section (Section 8, Conclusion & Limitations): if the learned ratio estimate is biased or if the downstream task truly requires sustained out-of-support behavior, ZOL can only trade off conservatism and reward.
>
> We view this as an inherent limitation of any test time adaptation (i.e LoLA, ReLA) method built on top of a fixed pretrained model, rather than a weakness unique to ZOL. However, **ZOL operates in offline setting without any additional samples from environment**.  Our contribution is not to remove representational error, but to use the pretrained FB model more reliably at inference time by selecting policy encoding latents in an occupancy-aware way. For simple conceptual example, we refer to our answer to Reviewer fvTf
>
> ---
>
> **Q3 - Why use the $\chi^2$ penalty?**
>
> We appreciate this question. The $\chi^2$ term is not an ad hoc variance penalty, but a principled occupancy regularizer. The variance penalty we use is $\hat{\chi^2}(z) = \frac{1}{B}\sum_{i=1}^B (w_i(z)-1)^2$. In Appendix C.5, Theorem C.2 rigorously proves that this penalty is mathematically equivalent to the $\chi^2$-divergence between the target policy occupancy and the behavior policy occupancy: $E_{d_\beta}[(w_\pi-1)^2] = \chi^2(d_\pi || d_\beta)$.
>
> This choice is standard in offline RL and density-ratio correction, and closely connected to methods such as DualDICE and OptiDICE, which also use divergence control to keep the target policy close to the dataset support. We agree that this motivation should be clearer in the main text, and we will move a concise summary of this result from the appendix into Section 3.
>
> ---
>
> **Q4 - Task Generalization**
>
> While deploying directly to real-world hardware is outside the scope of this paper, we strongly emphasize that our evaluation suite already includes highly complex simulated robotic tasks. We evaluate on OGBench, which includes challenging robotic manipulation environments such as ```cube-double```, ```cube-single```, and ```puzzle-3x3```. Also, our evaluation on the SMPL Humanoid involves controlling a $358$-dim state space with a $69$-dimensional action space. Extending this to real-world robots involves significant sim-to-real and hardware-specific challenges that are distinct from the core theoretical contribution of zero-shot off-policy adaptation.
>
> For discrete tasks, while our empirical focus is on continuous control (where representation learning is heavily studied), the theoretical derivations in Section 3 and App. C are action-space agnostic. In the discrete case, the FB representations learned are already close to optimal and thus the $z_{\text{init}}$ is already close to optimal and does not require further correction.
>
> We agree that results on real robots would be valuable future work, but they would also introduce sim-to-real and system-integration challenges beyond the main scope of this paper.
>
> ---
>
> **Q5 - Performance Gap**
>
> This is a good observation. We believe the smaller gain on AntMaze is due to the nature of these tasks: they primarily test long-horizon stitching, and the pretrained FB model already performs well there. In such cases, the inferred latent is less severely affected by support mismatch, so there is less room for ZOL to improve upon vanilla FB
>
> In contrast, ZOL is most helpful when the standard zero-shot latent is reward-aligned but occupancy-misaligned, which is more pronounced in tasks with stronger support shift. This is also why the gains are larger on several ExORL and OGBench settings than on some AntMaze variants.
>
> ---
>
> **Q6 - Table Typo**
>
> Thank you for catching this. For ```Antmaze-L-Stitch```, FB is the best-performing method in Table 1. We will correct this typo in the revision

---

### Official Review · Reviewer_D6HX · 2026-03-13

**Soundness:** 2
**Presentation:** 3
**Significance:** 3
**Originality:** 2
**Overall Recommendation:** 4
**Confidence:** 4

**Summary:**

This paper studies the off-policy learning issue in the task inference for a behavior foundation model for zero-shot RL based on the Forward-Backward (FB) representation framework. Specifically, to address the off-policy issue, the goal is to estimate the density ratio of stationary occupancy measures between the task-specific optimal policy and the underlying behavior policy. The authors found that under the FB decomposition, the density ratio is inherently captured by the F and B representations and can be used to infer an in-support latent task vector in the offline setting. Based on this finding, this paper proposes zero-shot off-policy learning (ZOL), which leverages this density ratio to search for a proper $z$ through gradient updates. The proposed inference method is then evaluated on a variety of benchmark tasks for zero-shot RL, including OGBench, ExORL, and AMASS.

**Compliance With Llm Reviewing Policy:**

Affirmed.

**Final Justification:**

Thanks to the authors for the rebuttal and the follow-up response.

This paper offers a novel approach (called ZOL) for integrating DICE-like off-policy learning with zero-shot RL based on the Forward-Backward (FB) representations. In the initial review, my main concerns are:

(1) Several parts of the proposed inference method seem problematic, especially the objective function for the search of the latent $z$ at test time.

(2) The proposed method is somewhat incremental given the FB literature since the density correction is applied only to those state-action pairs in the support of $d_{\beta}$ and that due to the regularization terms in $J_{\text{ret}}$ are not sufficiently motivated.

(3) Regarding the experiments, the empirical performance gain of ZOL appears not that significant, and the ablation study does not fully support the design choice.

At the discussion phase, the rebuttal adequately addressed the concern (1), but the other two concerns still remained.
In the authors’ follow-up response, the concern (2) was mitigated given the additional justification of the objective function and the regularization terms. The empirical evaluation has been also strengthened by the additional results.

After the discussions, the strength of this paper appears to outweigh the weaknesses. Hence, I have raised my score accordingly to reflect this.

**Key Questions For Authors:**

1. As mentioned above, some parts of the proposed inference method appear problematic, and the presentation about the proposed inference method is confusing. Can the authors clarify the actual objective used to search for $z$ at inference time and how $c_T$ can be a good initialization without resulting in out-of-distribution issues?

2. The empirical performance improvement of ZOL (e.g., Tables 1 and 2) appears not that significant compared to the baselines, e.g., vanilla FB and TD-JEPA. Using the search method of ZOL can sometimes even slightly hurt the performance of FB. Can the authors comment on this?

3. In Figure 6, the ablation study seems to contradict the proposed design since the performance is almost unaffected if the trust L2 or the $\chi^2$ coefficient is not included. It would be helpful to clarify this or better justify the design of the proposed objective function.


Additional comments:

- Lines 140-142, right column: “If the task reward can be represented linearly in the backward features, r(s, a) ≈ B(s, a)⊤z, then combining Eqs.(5)-(6) yields the familiar FB score…” This seems incorrect. In the FB framework, one does not need the linear realizability condition to get Q_r^{\pi_z} = F(s,a,z)^\top z.

- In Table 1: For the row of Antmaze-L-Stitch, FB is the best among the four methods (rather than ZOL).

**Limitations:**

The paper includes one short paragraph on the limitation of ZOL, i.e., ZOL inherits the representational bias of the learned FB factorization in the offline setting.

**Strengths And Weaknesses:**

Strengths:
1. This paper addresses an important issue in using FB representations for zero-shot RL.

2. The finding that Forward–Backward (FB) representations already implicitly contain the distribution-correction ratio is quite intriguing and can contribute to the understanding of the FB framework in general.

3. The empirical evaluation is quite thorough, covering a diverse suite of zero-shot adaptation tasks, including OGBench (goal-conditioned problems), ExORL (locomotation and maze), and AMASS.


Weakness:
1. Some parts of the proposed inference method appear problematic, and the presentation about the proposed inference method is also ambiguous and quite confusing.

-  In Lines 250-252, it is mentioned that “Using the FB approximation, our optimization objective function is given by (10). We perform a gradient ascent in the space of z.” However, it is unclear why maximizing the density ratio is preferred.
Intuitively, to mitigate OOD issue, one would prefer to have a smaller density ratio.

- In Lines 263-265, it is also mentioned that a penalized objective $\mathcal{J}\_{total}$ is used to search for $z$ at inference. However, $\mathcal{J}\_{total}$ seems disconnected from Equation (10).

- It is not immediately clear why the proposed inference method can work in a “zero-shot” manner. In Lines 264-266, one component in the regularized objective is $J\_{ret}$, which seems to be defined as the expected return of policy $\pi\_z$. However, in the offline RL setting, one typically can only access data from $d\_{\beta}$ induced by the behavior policy $\beta$ instead of $d\_{\pi\_z}$. It is unclear how $J\_{ret}$ can be evaluated in zero-shot without collecting data under $\pi\_z$.

- In Lines 243-247, $c\_T$ is calculated as in the standard FB framework and used as the initialization in the search of final $z$. However, this initialization seems to contradict the motivation of this work, which is to mitigate the out-of-distribution (OOD) issue incurred by the inferred latent $z$ originally calculated in the FB framework as in Equation (11). If one already starts from a $z$-induced policy that tends to query OOD regions compared to the offline dataset, it remains unclear why the proposed inference method can ultimately find a $z$ that is more within the support. Some clarifications would be helpful.

2. The proposed method is somewhat incremental given the existing FB literature. The main contribution is to provide a new heuristic to search for a potentially better latent $z$.

3. The empirical performance improvement of ZOL (e.g., Tables 1 and 2) appears not that significant compared to the baselines. For example, in Table 1, ZOL and the vanilla FB achieves comparable performance, and the improvement of ZOL is not statistically significant. Moreover, in some cases, the inference design of ZOL can hurt the performance compared to vanilla FB, e.g., Antmaze-L-Stitch and Antmaze-M-navigate. Regarding the experiments on OGBench, important zero-shot RL baselines like TD-JEPA and HILP are not included.

4. Some results in the ablation study seem to contradict the proposed design. For example, in Figure 6, the performance is almost unaffected if the trust L2 or the $lchi^2$ coefficient is not included. Then, it is unclear why these two terms are still needed.

---

> ### Author Rebuttal · Authors · 2026-03-28
>
> Thank you for your thoughtful questions. Below, we address each of your concerns.
>
> **Q1 - Inference objective and density-ratio role**
>
> **Eq 10 is not the objective being maximized**. It is our derived FB-based estimator of the stationary density ratio for the policy $\pi_z$ induced by latent $z$, i.e an occupancy-correction (DICE-like) term relative to the behavior distribution $d^\beta$
>
> We use this ratio to estimate the return of $\pi_z$ without online rollouts via regularized weighted-return objective:
>
> $$J_{\text{total}}(z)=J(\pi_z)-\lambda_1 \chi^2(z)-\lambda_2 \mathcal{L}_{\text{trust}}(z)$$
>
> where
>
> $$J_{\text{ret}}(\pi_z) = E_{(s,a,r) \sim d^{\beta}} \left[ w_{\pi_{z}/\beta}(s,a) \left( r(s,a) - \bar{r} \right) \right]$$
> Thus, we do not maximize the density ratio itself. We maximize an offline estimate of return with the ratio correcting the mismatch between $d^{\pi_z}$ and $d^\beta$.
>
> The method is conservative, evaluating only on $d^\beta$ support, with $w_{\pi_z/\beta}(s,a)=0$ when $d^\beta(s,a)=0$. For the exact ratio, $E_{d^\beta}[w_{\pi_z/\beta}]=1$, so the goal is not to make the ratio uniformly small, but to use it as a valid occupancy correction. Stability of search over $z$ is ensured via positivity, normalization, clipping, and trust-regions. We will revise Sec. 3.1 to make this clearer
>
> **Our correction-estimator derivation is inspired by DICE-style ideas, but is new in the zero-shot RL setting.**
>
> ---
> **Q2 - How Eq. (10) enters the regularized inference objective**
>
> The inference-time objective is
> $$J_{\text{total}}(z)=J(\pi_z)-\lambda_1 \chi^2(z)-\lambda_2 \mathcal{L}_{\text{trust}}(z)$$
>
> Eq. (10) is not a separate objective; it provides the density-ratio estimate used inside $J_{\text{ret}}$. For a candidate latent $z$ Eq. (10) defines $w_{\pi_z/\beta}(s,a)$, which is then used to weight rewards on the offline dataset. In short:
> $$z \xrightarrow{\text{Eq. (10)}} w_{\pi_z/\beta} \xrightarrow{\text{Return weighting}} J_{\mathrm{ret}}(z) \longrightarrow \mathcal{J}_{\mathrm{total}}(z).$$
>
> ---
> **Q3 - How is $J(\pi_z)$ evaluated in a zero-shot setting?**
>
> ZOL does not evaluate $J(\pi_z)$ using rollouts under $\pi_z$. Instead, it estimates the return of the candidate policy purely from offline data collected under $d^\beta$. For each candidate $z$, Eq. (10) provides an estimate of the stationary density ratio $w_{\pi_z/\beta}(s,a)$, which defines
> $$J_{\text{ret}}(\pi_z) = E_{(s,a,r) \sim d^{\beta}} \left[ w_{\pi_{z}/\beta}(s,a) \left( r(s,a) - \bar{r} \right) \right]$$
>
> Therefore, no samples from $d^{\pi_z}$ are required. In practice, Eq. (10) is computed from cached reset states and forward passes through the pretrained FB model, requiring no additional online interaction. ZOL remains zero-shot: test-time adaptation only searches over $z$ within the pretrained policy family
>
> ---
> **Q4 - Role of $z_{\text{init}}$ and OOD mitigation**
>
> There is no contradiction: standard FB inference provides only a reward-aligned initialization $z_{\text{init}}$, i.e., a task-relevant starting point in latent space, not the final solution. Because FB is learned off-policy under function approximation and limited data coverage, $z_{\text{init}}$ can be imperfect and support-biased.
>
> ZOL improves this initialization by optimizing $z$ under an occupancy-aware objective, so the final latent is selected based on the occupancy induced by the candidate policy rather than the vanilla FB rule alone. In this sense, $z_{\text{init}}$ provides task alignment, while the subsequent search corrects its support bias
>
> ---
> **Q5 (Technical Contribution) The main contribution is to provide a new heuristic.**
>
> We disagree that our method is heuristic. Our core contribution is theoretical: we formally link FB representations to stationary density ratios. This provides a principled explanation for zero-shot failures and a robust, offline method to improve pretrained FB models.
>
> ---
> **Q6 (Linearity assumption misunderstanding) "In the FB framework, one does not need the linear realizability condition"**
>
> We respectfully disagree. In the FB framework, zero-shot transfer to a new task requires reward linear realizability in the learned feature space. If the reward is represented as $r(s)=B(s)^\top z$ for $B: S \rightarrow R^d$, then (successor features [1, 2])
> $$Q_z^\pi(s, a) = E_\pi \left[ \sum_t \gamma^t B(s_t)^\top z \mid s, a \right] = E_\pi \left[ \sum_t \gamma^t B(s_t)^\top \mid s, a \right] z = F(s, a, \pi)^\top z$$
>
> [1] Touati et al: Learning One Representation to Optimize All Rewards
>
> [2] Agarwal et al: A unified framework for unsupervised reinforcement learning algorithms
>
> ---
> **Q7-Empirical Significance**
>
> The reviewer is correct that FB ($0.36 \pm 0.2$) outperformed ZOL ($0.3 \pm 0.1$) we will fix this in the paper. However, on ExORL, ZOL improves over FB on Cheetah Walk by +17.12% and Point Mass Top Left by +36%. OGBench is a sparse-reward, long-horizon benchmark with achieving a return (0.47 for ZOL vs 0.45 for FB).

---

> > ### Author Rebuttal · Reviewer_D6HX · 2026-04-03
> >
> > Thank you to the authors for the detailed response. The rebuttal has addressed many of the questions (in particular, Q1-Q4). I have some remaining concerns and follow-up comments / questions:
> >
> > **Regarding Q5:**
> > I agree that connecting the DICE idea with FB representations offer new and interesting insights into zero-shot RL. That said, the resulting ZOL algorithm still remains somewhat heuristic for two reasons: (1) Optimizing $J_{\text{ret}}$ does not necessarily maximize the total expected reward as the density correction is applied only to those state-action pairs in the support of $d_\beta$ (however, this is like an inherent limitation in offline RL). (2) The final objective function $J_{\text{total}}$ still deviates from $J_{\text{ret}}$, and it is not well motivated in theory how these regularizations would affect the resulting solution.
> >
> > **Regarding Q6:**
> > I understand the argument made in the paper and the rebuttal. By first setting $r$ to be in the span of $B$ (i.e., using $B$ as the features), then one can get the expression of $Q$ under FB. This (one-directional) connection from FB to successor features have been pointed out in both (Touati & Ollivier, 2021) and (Touati et al., 2023).
> >
> > However, this only shows that linear realizability is a sufficient condition (and does not imply that it is necessary). In my earlier comment, I wanted to point out that the description of the FB decomposition with a direct assumption that $r$ lies in the span of $B$ appears limited. Please correct me if I missed anything.
> >
> > **Some concerns in the review comments but are not addressed:**
> > - The empirical performance improvement of ZOL (e.g., Tables 1 and 2) seems not that statistically significant compared to the baselines (the gain seems much smaller than the standard deviation).
> > - In the experiments on OGBench, important zero-shot RL baselines like HILP and TD-JEPA are not included.
> > - Ablation: The performance appears almost unaffected if the trust L2 loss or the
> > $\chi^2$ term is not included.

---

> > > ### Author Response · Authors · 2026-04-05
> > >
> > > Dear Reviewer, thank you for your follow-up questions. We have provided clarifications below.
> > >
> > >
> > > **Regarding Q5:**  We would like to point out that we have included a detailed discussion on the theoretical effects and necessity of our regularizations in Appendix C.7 and we will elevate the relevant discussion from Appendix C to the main text.
> > >
> > > The $\chi^2$ penalty is not an arbitrary choice. As proven in Theorem C.2, our empirical variance penalty exactly equals the $\chi^2$-divergence between the target policy's stationary occupancy and the dataset's occupancy, defined mathematically as $\chi^2(d_\pi || d_\beta)$. As established in the DICE family of algorithms, explicitly penalizing this divergence binds the solution to policies whose state visitation remains supported by the dataset, which is one of the main practical motivations of our approach. The trust-region penalty, $\hat{\Omega}(z)$, allows us to control the drift from the initial $z$ because, while in zero-shot settings, we cannot tell how good the initial $z$ is. Additionally, drifting away from the initial value can affect the policy's specialization on the given problem.
> > >
> > > Therefore, optimizing our final objective keeps the entire solution anchored to the dataset and pretrained FB embeddings. We will clarify these derivations and motivations directly in the main algorithm section according to your comments
> > >
> > > **Regarding Q6:**  Thank you for the clarification. We agree that our wording in lines 140–142 was too strong. The linear realizability assumption $(r(s,a)\approx B(s,a)^\top z)$ is not required for the FB decomposition itself, nor for the general successor-measure identity $(Q_r^\pi(s,a)=\int M^\pi(s,a,s',a')r(s',a')ds'da')$. Rather, it is a sufficient condition under which Eqs. (5)–(6) recover the familiar successor-features / FB inner-product value function for particular policy $(Q_r^{\pi_z}(s,a)\approx F(s,a,z)^\top z)$. We thank the reviewer for pointing out this distinction and will make it explicit in the revision.
> > >
> > >
> > > **Answer to concerns:**
> > >
> > > - Our relative aggregate gains are competitive and align with the margins of progress established by recent state-of-the-art zero-shot RL papers (e.g., TD-JEPA, HILP, LoLA, ReLA). In Table 1, we reliably outperform both recent zero-shot methods (HILP, TD-JEPA; see table above) and online baselines (LoLA, ReLA). The same picture holds in Table 2, where ZOL yields significantly lower variance than the baselines. Importantly, we have **strict improvement over FB:** ZOL is built only on the pretrained FB backbone and improves upon it across every single evaluated task (16/16 on ExORL, Table 3).
> > >
> > > We would like to also emphasize that **our main contribution is in novel insight that zero-shot RL can be viewed as distribution correction estimation at test time. This connection was not previously investigated at all**. As for empirical performance: results on SMPL HumEnv (high-dim motion capture tasks) are much more pronouced (we refer to Table 7 in the Appendix)
> > >
> > > - We conducted additional experiments to answer this question. Our method performs better than HILP and TD-JEPA. **Moreover, both those methods (HILP/TD-JEPA) must be trained from scratch, which requires additional hyperparameter tuning per environment**. In contrary, **ZOL takes any pretrained FB model for this task and obtains higher performance without any additional updates, with no changes to the base model**. Which is much more appealing, given that there are already large pretrained BFMs (e.g Metamotivo)
> > >
> > >     | Task | HILP | TD-JEPA | ZOL |
> > >     |---|---:|---:|---:|
> > >     | antmaze-ln | 0.38 ± 0.1 | 0.42 ± 0.1 | **0.63 ± 0.1** |
> > >     | antmaze-ls | 0.1 ± 0.0 | **0.32 ± 0.1** | 0.30 ± 0.1 |
> > >     | antmaze-ms | 0.5± 0.1 | 0.55 ± 0.1 | **0.73 ± 0.2** |
> > >     | antmaze-me | 0.0 ± 0.0 | 0.2 ± 0.0 | **0.67 ± 0.3** |
> > >     | antmaze-mn | 0.83 ± 0.1 | 0.7 ± 0.1 | **0.91 ± 0.1** |
> > >     | cube-double | 0.0 ± 0.1 | 0.0 ± 0.0 | **0.01 ± 0.0** |
> > >     | cube-single | **0.44 ± 0.2** | 0.34 ± 0.0 | 0.34 ± 0.0 |
> > >     | puzzle-3x3 | 0.0 ± 0.0 | 0.11 ± 0.0 | **0.14 ± 0.1** |
> > >
> > > - We found that without these regularizations, performance drops, but including these regularizations positively affects performance, and the coefficients before them do not significantly affect it. The stability and lack of sensitivity to the regularization coefficients, we think this is a strength rather than a weakness of our method.
> > >
> > > **Concluding remarks**
> > > Thank you for your thoughtful feedback.
> > >  We would appreciate it if you could let us know whether our clarifications address your concerns.

---

### Decision · Program_Chairs · 2026-04-30

**Decision:**

Accept (regular)

**Comment:**

All reviewers agree that the paper addresses an important problem of FB-based offline zero-shot RL and that the proposed solution is interesting and novel. Their main concerns, mostly about clarity and limited experimental comparisons, were resolved by the authors during the rebuttal period. After discussion, all reviewers are positive about the paper and I do agree with their view. I am thus recommending acceptance and encourage the authors to revise the manuscript by integrating the reviewers' feedback and adding the new material provided in the rebuttal.